# Investigating university English as a foreign language instructors' implementations in teaching integral listening with speaking

Addisu Bogale Shago [1,2]*, Elias Woemego Bushisso[1], Taye Gebremariam Olamo[1]

1 Department of English Language and Literature, College of Social Sciences and Humanities, Hawassa University, Hawassa, Sidama Regional State, Ethiopia, 2 Department of English Language and Literature, College of Social Sciences and Humanities, Wolaita Sodo University, Wolaita, Southern Ethiopia Regional State, Ethiopia

* addisubogale2020@yahoo.com

## Abstract

Employing integral instruction of listening and speaking, and understanding their roles is significant for effective language teaching, developing learners' spoken and written proficiencies, and improving their achievement and motivation. However, there is limited prior research on EFL-integrated listening and speaking in public universities in Ethiopia. Investigating the integral teaching/learning of listening and speaking remains a research problem. Accordingly, the study aims to examine University English EFL instructors' implementations in teaching integral listening and speaking employing a concurrent mixed-methods design. A total of 252 sample respondents were involved in data collection. Comprehensive and systematic sampling techniques were used to select respondents using a 5-point Likert scale. Through purposive sampling, 12 instructors were identified for qualitative data collection employing semi-structured observations and interviews. Quantitative data analysis employed descriptive and inferential statistics run by SPSS version 20. Qualitative data were thematically analyzed utilizing NVIVO 12 Pro. The findings revealed a remarkable mismatch between EFL instructors' reported practice and the implementation of integral teaching of listening and speaking. Students' questionnaire data confirmed the mismatch between the results of the instructors' practice and implementation. The main themes predicted integral practice included using real-life listening materials, appropriate application of listening phases, and using the language features appropriately. The variance was indicated by the effect size with an eta-squared value of 75%. Qualitative findings supported that EFL teachers frequently relied on non-authentic listening materials. The study implied that EFL program development and curriculum reviews incorporate key factors influencing integral instruction of listening and speaking in the present study. Finally, the study provided recommendations for EFL teachers and the contexts beyond the present settings.

**Data availability statement:** The dataset is provided in figshare with DOI: 10.6084/m9.figshare.29291882.

**Funding:** The author(s) received no specific funding for this work.

**Competing interests:** The authors have declared that no competing interests exist.

## Introduction

Nowadays, communicating in English is crucial because it provides an opportunity for teaching/learning the language worldwide in academic and professional settings. English is a lingua franca [1]. Certainly, it helps as a language for international communication or a universal platform for science, education, business, and international relations [2,3]. Similarly, this position obliged people to use it as a unique language for wider communication or the language of the world, including African content [1,4].

African countries like Ethiopia use English as a medium of instruction and a language for scientific records in education settings [5]. Though Ethiopia is a multilingual nation, its education system offers English as a primary medium of instruction for higher education, including primary levels [6,7]. In Ethiopia, beyond being a medium of instruction, English is accepted as the language supporting the development of scientific research and internationalization. Nevertheless, the Ethiopian education system acknowledged the weaknesses of English language utilization in instruction and learning [8]. Consequently, the authors of the present research believe that the emphasis should be on utilizing English, i.e., the target language (TL), in an integrated skills approach in higher academic institutions, such as Ethiopian public universities. This is because employing language skills effectively in the classroom is the mission of Ethiopian education policy. This could minimize language-related hindrances to learning. The study used "*integral*" skills instead of "*integrated*" skills across the paper for consistency. The present study is aimed at "Investigating University English as a Foreign Language Instructors' Implementations in Teaching Integral Listening with Speaking."

Maintaining education goals requires an integral skills approach that incorporates language skills (listening, speaking, reading, and writing) as its natural occurrence. This process involves other language elements such as vocabulary, grammar, and pronunciation [9,10]. The skills integration requires effective listening comprehension using comprehensible input [11], and effective teaching and learning at universities need integral listening and speaking in real life. The reason is that listening and speaking are foundational for English language acquisition [12]. Additionally, integrating listening and speaking is considered a 21st-century skill that enhances language teaching/learning [13]. This indicates that the integration of listening and speaking (spoken medium) could have a dominant role over the written medium (reading and writing). The former maximizes the latter's learning and could help learners achieve language proficiency and successful learning outcomes [14].

English language teaching (ELT) might not be effective without comprehensive auditory development surrounding the reception of spoken and production of written mediums, which boosts understanding of spoken language [15]. Mere listening in an actual classroom might not increase meaningful language learning. It should incorporate integral utilization of language skills like speaking for spoken production and develop all-around language proficiencies [16]. Yet, integral listening and speaking are not intentionally considered in English lessons, which might affect the overall practice of English learning [17]. Lack of self-practice in addition to teacher-initiated classroom practice can significantly impact meaningful language learning [18].

Moreover, implementing an integral skills approach has gained little research attention in most countries, which could be due to various factors [19,20]. For example, challenges learners face in lexical competency and technology-integrated language learning and instruction [21], lack of deliberate teaching of vocabulary in the language classrooms [22], which could affect skills integration, gaps in blending learning to improve listening and grammar, realistic interaction using listening and speaking not only ears and mouth but also eyes which could include paralinguistic features [16], problems in testing and assessing listening and speaking skills due to the lack of vocabulary [22], factors related to strategy use for active listening frequently [23], metacognitive strategy utilization and foreign language (FL) anxiety [24], learning perception related to employing effective approach like CLT (communicative language teaching) that could improve listening and speaking [25].

In the context of Ethiopian higher education institutions, such problems appeared similar. For example, [26] and [27] state that researching and practicing the integral language skills approach received little or no attention in Ethiopia. Similarly, our experience revealed limited research on this approach, particularly listening and speaking. A few studies disclosed problems related to language policy awareness about meaningful language use [7] and the effective application of technology in English classes [28]. A spoken-based language pedagogy appeared negligible, and learners are unlikely to be part of sources that reflect on problems related to language teaching and learning processes. Yet, spoken-based language instruction develops learners' communicative competence [29].

Speaking fluency depends on listening comprehension, but it cannot be evidenced or controlled by an effective management system of language learning in a spoken context. This could result in anxiety in real language use in English language learning [14,30]. As a result, EFL learners face challenges in learning the language and misunderstanding the spoken messages, which could affect their quality of life and better opportunities in their educational world and future career [31]. This indicates that integral listening and speaking need a proper pedagogical approach to boost language learning in the classroom and calls for consistent practice using authentic listening materials [32]. Yet, in most countries, like Ethiopia, the practice is not frequent in language classrooms [26].

The teaching objectives of the course Communicative English Language Skills (I) emphasize integrating language skills. Thus, the lesson observation focused on the class periods of this course to explore the application of integral listening and speaking (the dependent variable). Successful EFL learning outcomes at universities could be achieved if the practice involves the influential factors (independent variables) as conceptualized in the present study. These included language features, employing authentic and familiar listening topics, implementing listening phases (using interactive bottom-up and top-down approaches), and motivational strategies, etc. could be grounded in language learning theories (cognitivism and constructivism) (see "Fig 1") [20,33–35]. They were categorized through theme-based approaches, i.e., a theory-based approach (based on reading literature) and a data-driven approach (emerging themes during data analysis), which our study considered as potential influencing factors (see "Fig 1"). We believe factors can advance the integrated instruction of listening and speaking in actual practice [34,36]. Despite its significant contribution to overall language learning development, the integral skills of listening and speaking instruction appeared to be disregarded in Ethiopian public universities.

Therefore, issues highlighted in this backgrounder prompted the current researchers to conduct the study: "Investigating University English as a Foreign Language Instructors' Implementations in Teaching Integral Listening with Speaking." A few previous studies investigated at national and international levels as mentioned earlier. The present study could bridge gaps and highlight significant insights for integrated listening and speaking instruction.

The present study is significant because it investigates integral listening and speaking to explore classroom practice and identify whether teacher response and student observation as a feedback align with each other. This research provides valuable insights into EFL teaching challenges in Ethiopian universities, making it relevant to curriculum developers, instructors, and policymakers. The findings could help as a blueprint for EFL teachers in Ethiopian public universities, including AMU and WCU, and beyond, who could fill gaps in teaching integrated listening and speaking and strengthen their teaching approach. This means that it contributes to enhancing the real practice of classroom pedagogy.

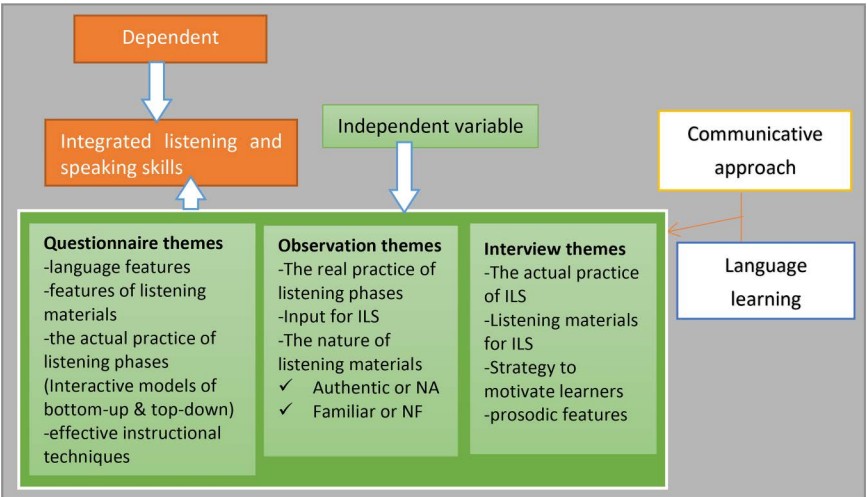

**Fig 1. Teaching practice of integral listening and speaking (conceptualized by the researchers).**

## Theoretical framework

The theoretical framework of this study relies on the concept of cognitivism and constructivism, language learning theories. These theories involve the communicative language method that enhances the integrated teaching practices of listening and speaking by considering the conceptualized thematic categories (independent variables) as illustrated in "Fig 1". These included language features, authentic listening material utilization, strategy instruction within the framework listening phases (interactive models of bottom-up and top-down), and utilizing appropriate teaching techniques such as scaffolding and identifying learning styles. These language learning theories can enhance EFL learners' ability to understand and engage learners in real-life communication through integrated listening and speaking [37]. Teaching listening is a psychological skill that involves cognitive abilities for meaningful communication [38,39]. It is believed that these theories support integrating listening and speaking to enhance EFL learners' language proficiency and emphasize utilizing authentic listening environments [13].

## Literature review

Integrated listening and speaking are developed through continuous classroom practices. This is because an integrated skills approach enhances EFL learners' aural-oral abilities and raises metacognitive awareness. For example, students who practiced regularly outperformed those using conventional methods [23]. Therefore, allocating significant classroom time to practice improves language proficiency [40]. Additionally, student-centered listening-speaking practices positively influence the language teaching-learning process and maximize English proficiency [20].

Knowledge features or elements, such as linguistic and discourse competencies, improve students' social interaction and support cognitive, metacognitive, and socio-affective instruction [41]. Discourse competence, instead, advances listening comprehension and speaking ability in EFL classes. However, discourse issues can negatively impact language teaching, especially listening and speaking [42]. Pragmatic knowledge improves learners' ability to understand spoken context and enhances meaningful language learning, maximizing speaking skills in the classroom [43,44]. However, pragmatic knowledge is inconsistently addressed in language classes [45], and its use in EFL classrooms is infrequent, with teachers often lacking practical implementation despite theoretical awareness [46].

Effective utilization of authentic listening materials boosts the teaching/learning of integral listening and speaking. Although employing appropriate authentic activities in integrated listening and speaking lessons is challenging for EFL

instructors, maximizing learning in real-life settings is crucial. Yet, EFL teachers will likely employ non-authentic listening tasks depending on the read-aloud technique to present listening and speaking activities [47]. For example, using audio-visual materials in teaching/learning listening maximizes learning motivation, and listening proficiency, and promotes speaking fluency in various contexts [48]. This approach involves utilizing social media to broaden language use and competence. However, EFL teachers must be prepared to incorporate real-life social media-based language learning in the classroom.

Before the listening phase, language teachers should integrate speaking and listening by pre-teaching difficult vocabulary and posing general questions to their students to activate learners' schemas. This is because pre-listening techniques have a valuable impact on listening comprehension [49]. Yet, like the present study's findings, the research suggests this stage is frequently overlooked in favor of listening and similarly provides little focus in post-listening [50]. In the listening phase, students listen, and teachers check them and ensure they understand by assigning relevant listening texts [51]. Most EFL teachers devote much time to this phase [50].

Explicit teaching of integral listening and speaking, or strategy instruction, can significantly influence listening performance as a concept-mapping technique [52,53]. Likewise, it is important in promoting learning, listening, and speaking integrally. Conversely, EFL teachers fail to implement actual strategy instruction in listening and speaking lessons because they perceive the task as demanding [33]. EFL instructors rarely employ strategy instruction in integrated listening and speaking classes [54]. The implementation was not evident except for positively believing in using interactive approaches, incorporating cognitive and metacognitive strategies, and bottom-up and top-down models that promote learning aural-oral skills in each listening phase [53,55,56]. With particular reference to the effective use of the metacognitive listening model (pre-listening, while listening, and post-listening), which helps learning become more valuable. It makes up a metacognitive model that helps us listen as input for effective speaking. Relevant listening exercises, listening stages support integrated teaching of listening and speaking practices by allowing teachers to activate, monitor, and assess their students' learning [57]. Effective listening activities in each phase enhance the learning process and boost EFL learners' active oral participation in group discussions [50]. Learners can excel when these stages are effectively utilized in the classroom [36]. Most language teachers do not emphasize aural-oral activities in each listening phase to enhance integrated listening-speaking lessons, often using only a few activities [58]. The teaching of integral listening and speaking is highly impacted by the lack of specific teaching techniques like CBI and TBI [59].

A hybrid model of content and task-based instructions is essential for skills integration. For instance, content-based instruction (CBI) integrates language teaching with subject matter content, allowing students to develop content knowledge and language skills through tasks focused on specific subjects. This approach naturally motivates students through cooperative tasks [60]. Task-based instruction (TBI), genre-based instruction, emphasizes real-world tasks and meaning-oriented activities. Rooted in communicative language teaching (CLT), TBI integrates all skills through meaningful tasks to achieve fluency and accuracy, enabling learners to explore and practice multiple skills [61]. Content-based instruction (CBI) is an effective technique for using language meaningfully and enhancing language learning performance. It incorporates relevant content to motivate EFL students, improving language proficiency and cognitive ability [62]. Task-based instruction (TBI) also significantly aids in teaching listening and speaking, integratively and independently. A mixed-methods study found TBI practical for EFL learners, enhancing context-sensitive teaching and improving listening comprehension [63,64].

Utilizing familiar topics in integral listening and speaking lessons enhances language learners' engagement, improves listening comprehension, and strengthens macro skill integration [65]. This is because common topics motivate learners and contribute significantly to developing spoken competence [66]. In this regard, culture-based familiar topics or incorporating diverse cultural topics enhance classroom conversation and speaking performance [67]. Yet, the use of unfamiliar topics in EFL instruction may hinder the integration of listening and speaking. This negatively impacts the pedagogical approach [68].

The scaffolding learning technique supports listening and speaking integration in EFL classrooms [69]. However, gaps were observed between teachers and students due to inadequate peer scaffolding and the predominance of teacher scaffolding in class discussions. EFL learners frequently showed passive involvement during listening exercises, resulting in a lack of active participation. This method wasn't specifically used, even though it was crucial for improving aural-oral integration [70].

The national and international studies underscore the substantial role of integrating listening and speaking for academic advancement and communication. However, the integral listening and speaking in learning/teaching remain unexplored, overlooked, and given little attention [71]. This problem seemed similar in Ethiopia, particularly in public universities where English as a foreign language (EFL) instructors often underscore theoretical aspects of skills integration but are less likely to implement practical instruction of listening integrated with speaking [26]. Likewise, teaching and learning listening in EFL classes often seemed neglected, focusing more on theory than practical application in real-life situations. The neglect and discrete teaching of speaking seemed to affect language learning and the overall performance of EFL. The reasons appeared to be poor listening, vocabulary, language background, EFL teachers' beliefs about English learning and teaching, effective pedagogical approaches like integrated instruction, and problems related to assessment [72]. Our preliminary observations and interviews at Wolita Sodo University and Hawassa University aligned with these findings. EFL students seemed to lack interest in learning the English language. The practice of listening integrated with speaking has not been widely implemented [27].

Teaching listening and speaking integrally seemed a difficult task for language teachers and students because of the nature of these skills [73,74]. For example, in spoken contexts, listening comprehension could be evaluated by speaking, but if listening is effectively utilized, the effective interaction would not be expected, which could erode learning. This is determined by learners' level of active engagement in listening to input, though it is highly personal. A lack of effective instructional techniques, like effective integrated language skills, listening and speaking skills, and difficulties in the formal examination of these skills, are perceived as obstacles [75].

Material, pedagogical, and cultural obstacles could hinder the application of integral teaching of listening and speaking [71]. Consequently, full EFL language learning and teaching needs a holistic strategy that develops integral listening and speaking. This is the rationale for teaching listening and speaking skills in an integrated way [76]. Ethiopian universities appeared not to recognize these oracy skills in an explicit mixed teaching approach in the classroom. It was observed that Ethiopian EFL students frequently work alone and receive little support from the language teachers to enhance their listening and speaking skills together [26].

Most works of literature mentioned above employed a single research approach and focused on a survey research approach, which could limit the quality of findings. Indeed, to minimize gaps, the present study employed a multiple-methods research approach, which integrates quantitative and qualitative research approaches. This could increase the comprehensive understanding of the problem under investigation. Similarly, the studies did not employ the explanatory and other variables conceptualized in this study. The present EFL context emphasizes the importance of inquiry concerning the integral skills approach, in which the curriculum sets its objectives as the foundation for language skills integration. However, the implementation of the integral teaching practice of listening and speaking was not evident during the preliminary observation. To address gaps, the present study aimed at "Investigating University English as a Foreign Language Instructors' Teaching of Integral Listening with Speaking" at Arba Minch and Wachemo Universities in Ethiopia. To attain the general purpose, the study established the following specific objectives.

1. To explore EFL instructors' implementation of integral listening and speaking skills instruction.

2. To examine the alignment of EFL learners' responses that reinforce instructors' implementation of integral listening and speaking skills instruction.

Generally, the study's conceptualized variables (independent variables) are demonstrated in "Fig 1", which shows the logical flow of the study variables. These variables could enhance the aural-oral skills of listening and speaking.

## Materials and methods

### Ethics statement

Initially, based on the ethical guidelines of Hawassa University, ethical approval was received from the standing committee of the College of Social Sciences and Humanities, which granted ethical clearance with the reference number CCH/123/2022 on June 9, 2022. This process was followed by getting permission from the Department of English Language and Literature (directed to the ethical committee) to clarify the purpose of the study. Then, the authors provided thorough information to the participants concerning the purpose of the research, clarified the confidentiality, anonymity, and closure of personal information, and restricted data access, which would not be disclosed to third parties. Similarly, unwilling participants could be withdrawn if they had insufficient time to participate in data collection. Finally, through in-depth discussion followed by reading and showing the letter that indicates obtaining permission from the department and the approval letter of the ethical committee, we received verbal informed consent from teacher participants. All these procedures were applied to student participants, which was clarified for PLOS ONE in response to how ethical issue of verbal consent in human research.

### Research design

The present study employed a mixed-methods research approach, particularly a concurrent mixed-methods research design within the umbrella of the pragmatism paradigm. Experts state that pragmatism and philosophical assumptions underpin the significance of ontology, epistemology, and axiology in the research process. Considering these, the current design integrates quantitative and qualitative methods to offer comprehensive viewpoints concerning the problem of the study [77]. It is important to collect and analyze data and merge findings concurrently. Epistemological issues in the study address theories of knowledge related to teaching integrated listening and speaking, guiding the investigation into the phenomena by exploring "what" and "how" questions. The study adheres to ethical research practices throughout data collection and presentation, aligning with the assumption of axiology. Questions about the nature and experiences of reality in teaching integrated listening and speaking reflect an ontological perspective [78,79] (see "Fig 2").

### The study setting and contexts

**Study setting.** The aim was to investigate University English as a Foreign Language Instructors' Implementations in Teaching of Integral Listening and Speaking. The study was conducted at Arba Minch University (AMU) and Wachemo University (WCU).

AMU was founded in 2004 and is located in Arba Minch City Administration at Gamo Zone in the South Ethiopia Regional State (newly structured in 2023). It is 505 km from Addis Ababa, Ethiopia, to the south [80]. Arba Minch is geographically located at $6^0 2'0''$ N, $37^0 33'0''$ E, and it is categorized under hotter air conditions, Kola [81]. AMU encompasses three technology institutes, including the Institute of Language and Culture Studies, six colleges, including the College of Social Sciences and Humanities, and four schools on different campuses.

WCU was founded in 2009 and is situated in Hossana City Administration at Hadiya Zone in the Central Ethiopia Regional State, which was newly structured in 2023. It is 230 km southwest of Addis Ababa, Ethiopia [82]. Hossana City is geographically situated at $7^0 32'59''$ N, $37^0 51'13''$ E. The city is surrounded by weather conditions of highlands, temperate midlands, and arid lowlands [83]. The university has more than 1479 academic staff and 2346 administrative staff. WCU encompasses seven colleges, including the College of Social Sciences and Humanities, where the English department is included. It launched another compass at Durame Town in Kembata Tembaro Zone. Within 73 UG and PG programs, it hosts more than 30,000 undergraduate and graduate students.

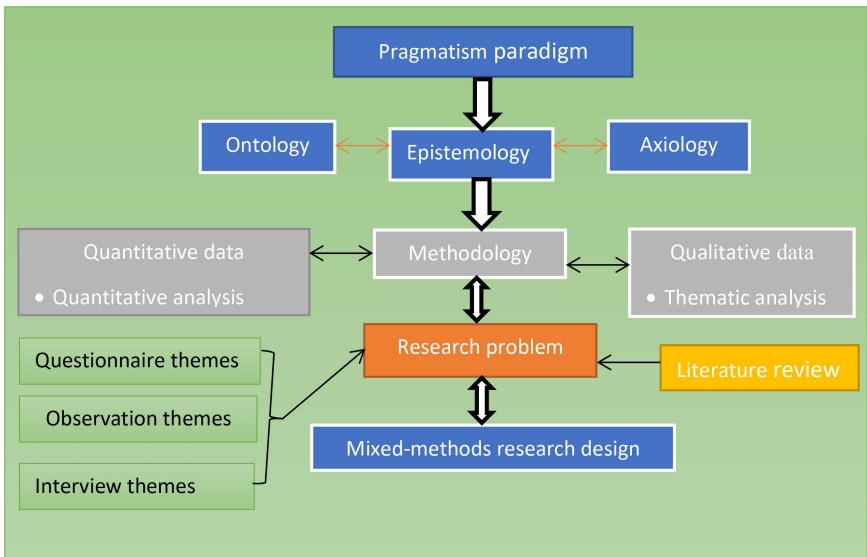

**Fig 2. A comprehensive approach to studying using mixed methodology.** Note: Thematic categories are illustrated in "Fig 1.".

**Study contexts.** The present study was conducted in the context of public universities (AMU and WCU) in Ethiopia, where English is taught as a foreign language (EFL) and used as a medium of instruction or target language (TL). Learning in TL can enhance EFL learners' language proficiency in listening and speaking skills. This study focuses on such integration because it is crucial for academic success at universities. Nonetheless, learners face difficulties acquiring proficiency in listening and speaking in an integrated manner. It seemed to cause problems in learning English language skills such as reading, writing, grammar, and vocabulary. The gaps in aural-oral skills proficiency could be due to learners' lack of exposure to authentic listening materials, regular practice, or limited opportunities for interactive listening, a lack of identifying learning styles, and using appropriate learning strategies, etc.

Communicative English Language Skills I (FLEn 1011), which is compulsory for first-year students, was the focus of the study's context. Its objectives appeared to develop learners' diverse perspectives on listening and speaking together if the practice is implemented as described in its objectives.

Data were collected within a month (May 01- May 30, 2023), from EFL instructors and students at Arba Minch and Wachemo Universities in Ethiopia.

## Sample size and sampling procedure

**Selection criteria for the study setting.** As mentioned earlier, this study was conducted at Arba Minch and Wachemo Universities, two of the 45 public universities in Ethiopia. We purposely selected these two settings. For example, we believed that, like the pilot study setting (HU), AMU and WCU have conducive and moderate weather conditions that might not affect the data collection. These Universities are well-established public Universities with relatively large EFL staff members in the Department of English Language and Literature. This helped the current researchers collect data and increase the sample size of the present study (n=92) at AMU and WCU from the pilot sample size (n=48). Similarly, the present study settings have similar teaching and learning environments, i.e., infrastructure and language resources to implement integral listening and speaking. This could help minimize variability in findings and increase the quality of the conclusion.

The gaps observed and identified in the present study settings were one of the purposes for sample selection. To indicate how these gaps were recognized, the authors have acquaintances and smooth communication with most staff

members of these universities. For instance, based on past exposure through workshops, symposiums, experience sharing in research and community services, and working together in previous careers at high schools and colleges. This helped the current researchers conduct informal preliminary discussions, observations, and interviews with a few staff members in each setting. Consequently, we understood the gaps in practicing integrated listening and speaking in the Communicative English Language Skills (I) course instruction.

Like the pilot study setting, the proximity of AMU and WCU facilitated the current researchers to collect data with little economic cost of time and money, and reduced fatigue.

## Sample size

EFL instructors assigned to teach the course Communicative English Language Skills (I) (FLEn 1011) in two selected settings participated in quantitative and qualitative data collection. For the survey, 92 instructors were selected using a comprehensive sampling technique. This technique ensures that all available participants in the study settings are included. It offers equal chances to individual participants in data collection [84].

A purposive sampling technique was utilized to select 12 teacher participants for qualitative data collection through interviews and observations. This helped the current researchers conduct an in-depth investigation on integral listening and speaking [84,85]. The inclusion criteria for categorizing interview participants were based on the level of education, years of experience, and current positions in the Department of English Language and Literature at two selected Universities. Principally, the position focused on the department head and the quality assurance coordinators. Therefore, participants in the interview were the key informants. Most participants in the interviews were high level academicians, i.e., PhD (associate professors) (n = 4), PhD (assistant professors) and department heads (n = 2), PhD (assistant professors) and ELIP (English Language Improvement Program) coordinators (n = 2), and PhD (assistant professors), quality assurance coordinators (n = 2), and MA (assistant professors) (n = 2). All interviewees (n = 12) have higher teaching experience, i.e., above 25 years of teaching experience in TEFL (Teaching English as a Foreign Language) (see "Table 1"). The English language departments of AMU and WCU disclosed the participants' teaching experience and level of education based on their profiles. They were willing to participate in the interview and observation. This helped collect data with little difficulty and ensured trustworthiness, and improved the quality of findings.

The lesson observation preceded the interviews, which helped avoid or minimize data contamination, and pre-information concerning the problem under investigation during the interview [86]. The lesson observation focused on the lessons of all interview participants. Therefore, the intentional (purposeful) choice of participants for qualitative data collection helped researchers elucidate ideas concerning the integrated teaching of listening and speaking in EFL classrooms. A scholar states that this technique helps provide a detailed understanding of the issue under investigation [87].

First-year (freshman) EFL students enrolled for the 2023 academic year and the Communicative English Language Skills (I) course participated in this study. Students placed in AMU and WCU (N = 266) were the population size. A total sample size (n = 160) was involved in survey data collection. How the sample size was determined? During planning data

**Table 1. Characteristics of interview participants.**

| No. | Education level | Rank | Gender | | Total | Years of Experience |
|-----|-----------------|------|--------|---|-------|---------------------|
| | | | Male | Female | | |
| 1 | PhD | Associate Professor | 4 | 0 | 4 | Above 25 |
| | | Assistant Professor | 5 | 1 | 6 | |
| 2 | MA | Assistant Professor | 2 | 0 | 2 | |
| Total | | | 11 | 1 | 12 | |

Note: Participants' specialization: TEFL (M = 9, F = 1 Total = 10); Linguistics (M = 1, F = 1 Total = 2).

collection, the researcher should define or estimate the required sample size to boost the quality of a study (either quantitative or qualitative), which is the first step [88].

Therefore, the present study utilized Yamane's formula. This helps calculate a sample size for a relatively small or exactly known population size [89]. The formula is:

$$n = \frac{N}{1 + N\ (e^2)};$$

Where:

"n" symbolizes the sample size

"N" symbolizes the population size, and

"e" symbolizes the level of precision (the margin of error)

The population of the present study is (N = 266), i.e., N of AMU is 146, and WCU is 120. The study measured a 95% confidence level (p-value = 0.05). Accordingly, the sample size (n) out of the total population (N) was calculated as:

$$n = \frac{266}{1 + 266(0.05^2)}$$

$$n = \frac{266}{1 + (266\ (0.00255^2))}$$

$$n = \frac{266}{1 + (266\ (0.665))}$$

$$n = \frac{266}{1..665} = 159.759759 \approx 160$$

The sample size was allocated to each research setting using a proportional allocation method. For example, the n of AMU was $\frac{146}{266 x 160}$ = 88, whereas the n of WCU was $\frac{120}{266 x 160}$ = 72

A systematic sampling technique was utilized to create a representative sample of student respondents at AMU and WCU. This method ensures equitable recruitment across the entire student population for the study. The random points were calculated using the sampling interval formula ($k^{th}$ value = $\frac{N}{n}$), where "k" symbolizes the interval value, "N" symbolizes the total student population, and n denotes the sample size. From the N of 266 student population at two selected settings, to see the random interval point, i.e., $\frac{266}{160}$ = 1.6625 $\approx$ 2. Therefore, a random point was included for every 2nd, 4th, 6th, etc. intervals to choose student respondents until it reached the target sample size in each university from the list [90].

### Instruments

This study employed questionnaires, personal observations, and interviews. This helped to increase the data richness and quality of findings.

**Questionnaires.** The questionnaire items were adapted to practice teaching and learning integrated listening and speaking. This instrument employed a 5-point Likert scale questionnaire to measure the level of actual teaching practice of integrated listening and speaking in the EFL classroom. These scales include "Often" (5), "Always" (4), "Sometimes" (3), "Rarely" (2), and "Never" (1). The questionnaire included 22 closed-ended items for teacher and student respondents (11 items each) [91]. The items were prepared in two forms. The first form was targeted to instructors to fill out using

Google Forms. Another form was intended to administer a questionnaire for EFL students because the researchers couldn't access the email addresses of all student respondents. This was because learners witnessed that they come from diverse backgrounds, such as rural, urban, and some remote corners of the country. Indeed, most students were not exposed to utilizing technology and accessing email addresses, and were unwilling to complete surveys online. Yet, a few students accessed the email addresses. This disparity could affect the consistency of data collection. Therefore, for effective management, the researchers believed that administering the questionnaire in person could increase validity and minimize the missing values of the quantitative data. The questionnaire data collected from the students was intended to triangulate with the instructors' surveys.

The questionnaire was piloted at Hawassa University, with instructors (n = 48), to examine internal consistency using Cronbach's alpha. Its score was 0.85. It was reliable. To ensure the quality of the questionnaire in other contexts for the main study, the reliability was re-tested through Cronbach's alpha, and its score is equal to 0.930. Further, the reliability of each sub-dimension of the questionnaire that enhances integrated listening and speaking was calculated and reported as Theme 1 (utilizing language features and other variables) = 0.860, Theme 2 (features of listening materials) = 0.870, Theme 3 (the implementation of listening phases) = 0.882, and Theme 4 (effective instructional techniques) = 0.850. This confirms that the questionnaires were reliable and ready to collect data for the main study. The study revealed that in addition to testing reliability in the pilot study, testing the reliability in the post-piloting, or in the main research (large-scale study), increases the quality of instruments [92]. So, the quality of the instruments was achieved.

The validity was ensured in both phases (pilot and main). The expert judgment (supervisors and randomly selected four EFL instructors) (2 from AMU and 2 from WCU) in the field of study (ELT) ensured the validity of instruments (question-naires, interviews, and observations), particularly content and face validity, before self-administering the questionnaires. This helped the researchers to validate the content, i.e., what we wanted to measure and the face value of the instru-ment, so that it would appear well structured and attract audiences in a good organization. Accordingly, they corrected the instructions, difficulty level, clarity, and scale order, and reduced the number of pages of the instruments to collect data. Accordingly, potentially problematic two items from the questionnaire and two irrelevant or overlapped items from the observation and interview were discarded.

The study tested the assumptions of normality using checkmark histograms, normal probability plots, and scatter plots, which showed a normal distribution of data [93].

Likewise, multicollinearity statistics, such as tolerance and variance inflation factor (VIF), were tested and indicated lower values (<10 and < 2), respectively. These issues can affect the coefficient of determination (R-squared) if it exceeds these numerical values. However, in the present study, VIF and tolerance values were shown to be expected standard values (see "Table 4"). This indicates no potential problems with multicollinearity among independent or explanatory vari-ables. This could improve the regression model. Therefore, this model is accurate, reliable, and stable [94].

**Observation.** The personal non-participant observation explored the practical aspects of integrated listening and speaking in the classroom. It offered deeper insights into teaching practices and addressed gaps not covered by other methods, like questionnaires and interviews [95]. A few structured items (24) under four theme categories were analyzed qualitatively using PLATO scales. The analysis was based on counting the frequency of 12 lesson observations, followed by transcription of verbatim, which helped interpret qualitatively. The observation process incorporated PLATO observation scales rather than employing interrater reliability. The observations were not conducted by involving multiple observers. Conversely, it was conducted by a single observer (the corresponding author), a PhD candidate (investigator) in this study, and supervised by the co-authors. The behaviors in the classroom were managed by the first investigator rather than involving multiple observers. Consequently, we did not involve multiple observers to assess interrater reliability. Studies indicate that inter-rater reliabilities can be conducted if situations are observed by more than one observer [96]. Inter-coder or interrater reliability is calculated based on the extent to which two or more coders agree on the codes applied to a fixed set of units in qualitative data based on the nature of the research [97].

Three open-ended items were adapted in line with skills integration and qualitatively analyzed using NVIVO [98]. These themes include pre-listening, listening, post-listening, and input for integrated listening and speaking. Additionally, three open-ended items were categorized under three themes, including the implementation of listening phases, the nature of listening material, and the characteristics of listening topics. A total of 27 items were utilized to categorize the seven themes of the observation and were adapted in line with skills integration [98] (see "S4 Appendix").

### Interview

The present study employed qualitative data collection instruments like interviews and observations. These instruments helped complement quantitative findings. Semi-structured face-to-face interviews were used to collect data for an in-depth understanding linked to the integrated teaching practice of listening and speaking. The interviews followed an interview protocol based on thematic categories focusing on the main question items of the interview, followed by some probing items [85] (see "S3 Appendix"). The three main categories include the actual practice of integrated listening and speaking, listening materials for integrated listening and speaking, and strategies to motivate learners in integration. This approach elicited detailed insights into teaching practice in an integrated skills approach involving EFL listening and speaking.

To increase the trustworthiness of qualitative data, the study employed all necessary steps from data collection using interviews and observations to reporting findings. To boost the reliability of the instruments and related findings, the present research used a consistent approach with a few qualitative research procedures [77]. The validity of the interview and observation guides was established through expert judgment, similar to questionnaires. These instruments were written in English (the target language). This is because it is the medium of instruction in higher education institutions in Ethiopia, like universities. Similarly, all university instructors are expected to use the target language. Therefore, the interviews were conducted in the target language.

### Procedures of data collection

The data were piloted for validity and reliability. Following a pilot study, the researchers collected data for the main study from May 1–30, 2023. By showing an approval letter to the Departments of English Language and Literature of AMU and WCU, consent was obtained from the AMU and WCU English departments. Following this, we collected instructors' addresses, such as phone numbers to communicate with participants about data collection, and emails to attach the survey online. Accordingly, the activities were scheduled for four weeks, which helped effective time and data management. Initially (the first week), lesson observations were conducted in the classrooms. In this stage, the researchers familiarized themselves and built interactions with participants to collect data from face-to-face interviews in the second week.

Pre-interview observation is essential because it can provide significant insight into the problem and improve the interaction with the participants, facilitating successful accomplishment in the interview process. For example, conducting an observation before an interview could provide input for researchers. This increases the researcher's understanding of the participants in their natural setting and could make the interview meaningful [86]. This increases the systematic investigation of the problem [87].

The principal investigator of the current study (author 1, a PhD candidate) conducted lesson observations and interviews. The duration of 12 observation sessions took an average of 1 hour each. The duration of 12 interviews took an average of 40 minutes. Interviews and observations were audio recorded, and field notes were used to interpret participants' paralinguistic features, as they could not be captured by audio recording. The recording assisted the researchers in minimizing forgetting and increasing the accuracy of data collection. The qualitative data were immediately transcribed verbatim for data analysis. For effective data collection from participants, the interviewer clarified the interview guides if interviewees did not understand the main themes of the interview items. The interviewer let the participants freely provide their opinions regarding integrated skills integration. If the participants felt the questions were difficult, the interviewer (a PhD candidate) would immediately adapt or rephrase the questions, as the semi-structured interview provides the room to

do so. Then, the interviewer informed the adaptation of items to the supervisors (co-authors in this article) as it is necessary. This helped to make improvements, i.e., if not relevant, discard the interview questions to enhance the validity of findings.

The present study employed a concurrent mixed-methods design, providing room for side-by-side data collection. Thus, the quantitative data collection was immediately followed by the qualitative data collection to complete the planned study period. Therefore, the surveys were sent to EFL instructors via Google Forms, and all teacher participants (n = 92) filled out the questionnaire. Consequently, no missing values were recorded. Then, the raw data of the questionnaire online (Google Forms) were downloaded to an Excel sheet and entered into SPSS version 20. Conversely, the questionnaire was administered to student respondents (n = 160) in hard copy and then entered into the same SPSS for data analysis.

### Data analysis

**Quantitative data analysis.** Quantitative data from EFL instructors and students were analyzed using SPSS version 20, incorporating descriptive and inferential statistics (multiple regressions). These methods assessed respondent characteristics and variable productivity. The enter method categorizes themes based on related items for skills integration. The items were identified into four distinct categories based on data exploration. For example, Theme 1 (utilizing language features) included three items (items 17, 23, and 24), Theme 2 (effective utilization of authentic listening materials) included four items (items 18, and 19), Theme 3 (the implementation of listening phases) addressed item 26, and Theme 4 (effective instructional techniques) incorporated five items (items 20, 21, 22, 25, 27). It is noted that sorting variables into themes is valuable for non-experimental studies to indicate predictive levels, like the present study, which did not manipulate variables [99].

The thematic analysis used the enter method using different predictor selection techniques and has different steps in predictor selection in the regression model. These included stepwise, ridge regression, and lasso (least absolute shrinkage and selection operation) model selection techniques. However, each method has its strengths and weaknesses. The stepwise predictor selection technique helps combine the addition (forward selection) [100] and the deletion (backward selection) of variables based on their importance. Thus, this technique is flexible and can balance complexity in building the model [101]. Ridge regression technique facilitates optimizing higher coefficients and reducing overfitting in the regression model. It deals with the multicollinearity issue. Further, Lasso helps balance the overfitting of the model in regression analysis and shrinks the coefficients in the model [101,102]. Therefore, these techniques are significant for the present study and are utilized to select strong predictors of teaching integral listening and speaking. These predictors included utilizing language features, the features of listening materials, and the implementation of listening phases, which were included in the model. A weak predictor (Theme 4), i.e., using effective instructional techniques, was removed from the regression model, which was calculated for teacher and student respondents for comparison.

Though the result of regression statistics shows relatively lower value of adjusted $R^2$ (.288, i.e., 28.8%) (See "Table 3"), the confidence level or interval (in this study $p < 0.05$, i.e., 95%) could be considered for goodness of fit (see "the coefficients$^a$" in "Table 4"). This means that the lower the value of the adjusted R-Square, the higher the increase in confounding factors, and it can't be controlled by the researchers. Similarly, in such cases, issues like VIF could be considered if it is not violated, i.e., its value should be less than 2, like in the present study [103]. A smaller sample size, like the present study, can affect the adjusted $R^2$ value in the research process [104].

**Qualitative data. Coding frame for qualitative data analysis:** The coding frame helps analyze and later discuss it accordingly. In the present study, special attention was paid to connecting variables (dependent and independent) that were theory- and data-driven as conceptualized in this study. It combined these approaches to the coding framework of qualitative data. Thus, the process passed two phases. This helped to conceptualize the variables under study [105,106]. The study states that a research process should indicate the relationship among variables that could be based on an effective coding frame [99].

Initially, the study used theory-driven themes based on reading-related (existing) literature to decide or establish components (dependent and independent variables). This means researchers pre-determined a few codes, i.e., a coding frame based on literature related to integral listening and speaking. This way of coding is called deductive coding [107,108]. Accordingly, the coding frame specifically focuses on independent variables that could boost the teaching practice of this integration. For example, "language features", "features of listening materials", "the actual practice of listening phases", "effective instructional techniques", and "Input for ILS". Next, or phase 2, a data-driven approach helped identify emerging themes (coding frame) during data analysis and coding. This indicative approach, where data or themes are explored without prior determination (beyond the theoretical theme), i.e., an emerging coding frame based on many children's nodes [109]. Based on this method, two themes emerged. For instance, "strategy to motivate learners" and prosodic features". Generally, the study employed the three-cycle coding methods of [108] (See "S5 Appendix").

**Qualitative data analysis.** Qualitative data were thematically analyzed, which involved careful coding thematically [110]. The closed-ended observation data were analyzed using PLATO scales ranging from "*Not evident*" to "*Evident,*" and each scale description is provided (see "S4 Appendix"). This assisted in quantifying the data within the two themes ("The real practice of listening phases, i.e., pre-, while, and post-listening)" (Table 8) and "Input for integrated listening and speaking instruction") ("Table 9") before qualitative interpretation based on numbers [111]. Open-ended observation data supplemented by three themes ("S4 Appendix") were analyzed using NVIVO 12 Pro. Similarly, NVIVO software helped to analyze interview data in three themes: "The actual practice or implementation of integrated listening and speaking", "Listening materials for integrated listening and speaking", and "Strategy to motivate learners in integration" (see "Table 10"). The theming technique involved the theoretical basis (highly emphasized) and data-driven (emerging), connected to integrated listening and speaking skills. The data analysis of this study employed [77] seven-step qualitative data analysis process (see "Fig 3").

## Results

After analyzing data based on 14 major leading categories (eight categories for quantitative data and six categories for qualitative data), the results are presented precisely in tabular form below, followed by clear descriptions.

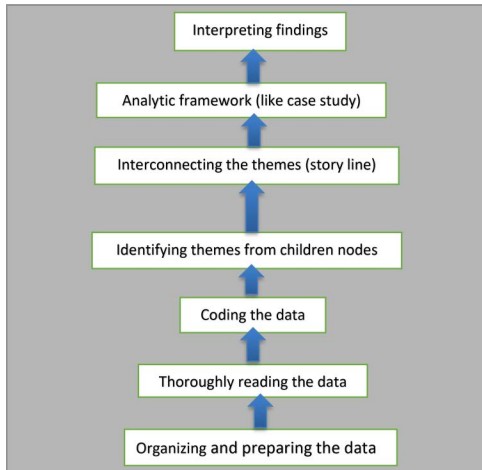

**Fig 3. Steps in the qualitative data analysis.**

**Table 2. The mean (M) and standard deviations (SD) of instructors' practices (N = 92).**

| No. | Statement | M | SD |
|---|---|---|---|
| 17 | I continuously practice teaching listening in integration with speaking for effective EFL learning. | 3.74 | 1.047 |
| 18 | I use authentic listening materials like audio and video to practice teaching listening skills in integration with speaking skills. | 3.68 | 1.058 |
| 19 | I utilize familiar topics to practice teaching listening skills in integration with speaking skills. | 3.93 | .875 |
| 20 | I motivate learners to reflect on listening lessons to review listening activities to teach listening in integration with speaking skills. | 4.03 | .762 |
| 21 | I apply the conversational dialogue to teaching listening skills in integration with speaking skills. | 3.28 | 1.170 |
| 22 | I provide feedback to learners in teaching listening skills in integration with speaking skills. | 3.54 | .954 |
| 23 | I employ linguistic, discourse, pragmatic, etc. information to practice teaching listening skills in integration with speaking skills. | 3.41 | .939 |
| 24 | I explicitly teach listening skills in integration with speaking skills to develop learners' listening competencies. | 3.70 | .980 |
| 25 | I teach listening skills in integration with speaking skills by implementing task-based instruction, incorporating speaking content into listening lessons. | 3.67 | .939 |
| 26 | I effectively implement pre-, while, and post-listening stages to teach listening skills in integration with speaking skills. | 3.68 | 1.099 |
| 27 | I employ techniques like scaffolding note-taking, opinion sharing, etc., to teach listening skills in integration with speaking skills. | 3.77 | .927 |

https://doi.org/10.1371/journal.pone.0327029.t02

## Descriptive statistics

"Table 2" shows that most items, except for items 21 (M = 3.28; SD = 1.170) and 23 (M = 3.41; SD = 0.939), have mean values exceeding 3.5. The results indicate that 7 of 11 items have lower standard deviations, suggesting less response unevenness. This implies a moderate mean value and lower standard deviation, indicating that instructors likely practice integrated listening and speaking consistently in EFL classrooms. (See "S1 Appendix").

## Inferential statistics

"Table 3" shows a moderate positive correlation (R = 0.565) between independent variables (IV) and integral listening and speaking (DV). The R² value of 0.319 indicates that 31.9% of the variance is explained by the IV, with a standard error of 0.49447, suggesting accurate predictions.

"Table 4" shows the coefficients for the predictor (classroom practice). Themes 1, 2, and 3 have positive and significant relationships with the dependent variable (integrated listening and speaking) at p < 0.05. Conversely, Theme 4 is insignificant at p > 0.05. Themes 1 (utilizing language features) (B = 0.351; Beta = 0.31.6), 2 (effective utilization of authentic listening materials (B = 0.308; Beta = 0.250), and 3 (the actual implementation] of listening phases (B = 0.204; Beta = 0.203) significantly predict integrated listening and speaking. Yet, Theme 4 (utilizing appropriate instructional techniques, B = 0.102) was removed from the model because of a lower predictive power of DV during predictor selection. Themes 1 and 2 show the strongest prediction, followed by Theme 3.

Tolerance values (TL) > 0.1 and variance of inflation factor (VIF) < 10 indicate no potential multicollinearity concerns. This means that there is no correlation among independent variables.

"Table 5" shows that most items (9) have low mean values, with only two items above 2. The average mean is 2.01, with a standard deviation of 0.755. This indicates that instructors rarely integrate listening with speaking in EFL classrooms, contrary to "Table 4" findings. (See "S3 Appendix").

"Table 6" shows a weak correlation between dependent and independent variables (R² = 0.133), explaining only 1.8% of the variance. The model has a weak fit, with a low standard error (0.43090) indicating minimal variability in responses.

"Table 7" shows that all practice themes have statistically insignificant (p > 0.05) and no significant relationship with teaching integrated listening and speaking. The small B values indicate that the predictors (themes) have no significant effect on the dependent variable.

The tolerance values are greater than 0.1, and the variance of inflation factor (VIF) values are less than 10. This suggests no multicollinearity issue.

**Table 3. Model summary[b] of instructors' practice.**

| Model 1 | R | R² | Adjusted R² | Std. Er. Est.=Std. |
|---|---|---|---|---|
| | .565[a] | .319 | .288 | .49447 |

Note: a. Predictors: (Constant), Theme 1, Theme 2, Theme 3, Theme 4 (deleted during the predictor selection process) b. Dependent Variable, Std. Er. Est.=Std. Error of the Estimate. Yet, Theme 4 was removed from the model for its weak prediction.

**Table 4. The coefficients[a] of instructors' practices (N=92).**

| Model 1 | USTCo | | STCo | T | Sig. | Correlations | | | CoSta | |
|---|---|---|---|---|---|---|---|---|---|---|
| | B | Std. E | Beta | | | Z | Partial | Part | TL | VIF |
| (Constant) | 1.332 | .455 | | 2.928 | .004 | | | | | |
| Theme 1 | .351 | .131 | .316 | 2.674 | .009 | .478 | .276 | .236 | .560 | 1.785 |
| Theme 2 | .308 | .124 | .250 | 2.481 | .015 | .422 | .257 | .219 | .769 | 1.301 |
| Theme 3 | .204 | .092 | .203 | 2.219 | .029 | .310 | .231 | .196 | .936 | 1.068 |

Note: a. Dependent Variable (DV); N (total sample population); USTCo (Unstandardized Coefficients); STCO (Standardized Coefficients); Std. E (standard error); ZO (Zero-order); TL (Tolerance); CoSta (Collinearity Statistics).

**Table 5. The learning practice of listening integrated with speaking (N=160).**

| No. | Statement | M | SD |
|---|---|---|---|
| 1 | My teacher continuously teaches listening in integration with speaking for effective learning. | 1.81 | .843 |
| 2 | My instructor uses authentic listening materials like audio and video to practice teaching listening skills in integration with speaking skills. | 1.73 | .642 |
| 3 | My instructor utilizes familiar topics to teach listening skills in integration with speaking skills. | 1.74 | .566 |
| 4 | My instructor motivates me to reflect on listening lessons and review listening activities to teach listening in integration with speaking skills. | 1.76 | .671 |
| 5 | My instructor applies the conversational dialogue to teaching listening skills in integration with speaking skills. | 1.66 | .643 |
| 6 | My instructor provides feedback to me on teaching listening skills in integration with speaking skills. | 1.76 | .781 |
| 7 | My instructor employs linguistic, discourse, pragmatic, etc. information to practice teaching listening skills in integration with speaking skills. | 2.02 | 1.073 |
| 8 | My instructor explicitly (clearly) teaches listening skills in integration with speaking skills to develop my listening competencies. | 1.88 | .747 |
| 9 | My instructor teaches listening in integration with speaking through task-based instruction incorporating speaking content into listening lessons. | 1.76 | .668 |
| 10 | My instructor effectively implements pre-, while, and post-listening stages to teach listening skills in integration with speaking skills. | 1.82 | .768 |
| 11 | My instructor employs techniques like scaffolding note-taking, opinion sharing, etc. to teach listening skills in integration with speaking. | 4.27 | .909 |

**Table 6. Model summary[b] of students' practice (N=160).**

| Model 1 | R | R² | Adjusted R² | Std. Er. Est. |
|---|---|---|---|---|
| | .133[a] | .018 | −.008 | .43194 |

Note: a. Predictors: (Constant), PRS; b. Dependent Variable: Std. Er. Est.=Std. Error of the Estimate.

**Table 7. The coefficients[a] of students' practices (N = 160).**

| Model 1 1 | USTCo | | STCo | T | Sig. | Correlations | | | CoSta | |
|---|---|---|---|---|---|---|---|---|---|---|
| | B | Std. E | Beta | | | Z | Parti-al | Part | TL | VIF |
| (Constant) | 4.525 | .376 | | 12.038 | .000 | | | | | |
| Theme 1 | −.032 | .092 | −.028 | −.344 | .731 | −.032 | −.028 | −.027 | .955 | 1.048 |
| Theme 2 | −.126 | .093 | −.109 | −1.359 | .176 | −.099 | −.109 | −.108 | .985 | 1.015 |
| Theme 3 | .019 | .065 | .023 | .284 | .777 | .016 | .023 | .023 | .969 | 1.032 |

Note: a. Dependent Variable (DV); N (total sample population); USTCo (Unstandardized Coefficients); STCO (Standardized Coefficients); Std. E (standard error); ZO (Zero-order); TL (Tolerance); CoSta (Collinearity Statistics).

"Table 8" shows ANOVA to examine differences across groups (teacher and student). The table illustrates a statistically significant difference in practice or implementation of integral listening and speaking between the two groups at (F = 752.964, p = 0.000 < 0.05 (1-tailed), eta-squared = 0.750, explaining a very large effect. The table illustrates the result of group statistics of practice, where the sum of squares = 139.065 with 1 degree of freedom (df), demonstrated in a mean square of 139.065.

From the "Table 8" it is possible to calculate eta-squared using the mean differences in ANOVA employing the formula $\eta^2 = \frac{SSB}{TSS}$. Where $\eta^2$ is eta-squared, SSB is the between-group sum of squares, and TSS is the total sum of squares. Therefore, the eta-squared is calculated as:

$$\eta^2 = \frac{139.065}{185.237} \approx 0.750.$$

This indicates that 75% variance in the integral teaching of listening and speaking is explained by the independent variables (actual implementation in the classroom).

Just as Cohen's d (d), or Cohen's effect size (f), which represents the standard deviation of standardized group means, can be calculated from eta-squared using the formula $d = \sqrt{\frac{\eta^2}{1-\eta^2}}$. Where d is Cohen's d or effect size, and $\eta^2$ is eta-squared. Accordingly,

$$d = \sqrt{\frac{0.750}{1-0.750}}$$

$$d = \sqrt{\frac{0.750}{0.250}}$$

$$d = \sqrt{3} \approx 1.732$$

**Table 8. Inferential group comparison of practice.**

| One-Way ANOVA | | | | | |
|---|---|---|---|---|---|
| | Sum of squares | df | Mean square | F | Significance |
| Between groups | 139.065 | 1 | 139.065 | 752.964 | .000 |
| Within groups | 46.172 | 250 | .185 | | |
| Total | 185.237 | 251 | | | |

## Qualitative findings

**Observation data.** Findings from this tool indicate the actual classroom practices through personal observation.

"Table 9" shows the thematic analysis of three themes and sub-themes. In Category 1, pre-listening is observed 17 times, less frequently than while-listening (35 times) and post-listening (20 times). Most participants (n = 8) do explicitly teach familiar vocabulary, though a few (n = 4) include terms like "career" and "profession." Pre-listening and post-listening stages are less emphasized compared to while listening. In the while-listening stage, most participants (n = 8) read aloud, while a few (n = 4) use authentic materials. In post-listening, most classes (n = 3) showed passive participation, with few students responding individually.

In Category 2 of "Table 9", only a few participants (n = 3) use appropriate authentic listening materials, while the majority (n = 9) do not, with only 9 references to such materials in observations. For instance, T2, T3, and T5 use authentic audio materials via personal computers, motivating students. In contrast, most participants (n = 9) predominantly use non-authentic materials, referenced 45 times in NVIVO, with similar findings from interviews. In Category 3, most participants (n = 8) used unfamiliar topics for teaching integrated listening and speaking, observed 38 times, whereas familiar topics were used only 13 times by T2, T7, T8, and T10.

"Table 10" summarizes the findings on the frequency of teaching practices. According to the findings, pre-listening was rarely evident, with a frequency of 24 times, but was only observed 7 times as evident. Post-listening was similarly infrequent, evident 18 times and not evident 16 times. In contrast, listening was frequently observed, with 41 instances somewhat evident and 30 evident. The results indicate minimal attention to pre- and post-listening stages, with appropriate listening input also rarely evident, observed 18 and 16 times, respectively. (See "S4 Appendix").

## Findings of the interview data

"Table 11" identifies three major categories with sub-categories. (See "S2 Appendix and S6 Appendix" for details). Category 1 shows that EFL instructors do not deliberately integrate listening and speaking in their classrooms. Interview findings support this, with 46 redundancies highlighting difficulties in integration. For instance, T1 and T5 mentioned, "It is difficult to apply the integration of listening and speaking as listening is an ignored skill, and no attention is given to integration," and T7 stated, "I do not integrate deliberately." A lack of suitable tasks contributes to this issue, as noted by T5 and T8, who observed that instructors skip listening sections due to inadequate tasks and resources.

**Table 9. Findings of open-ended observation (thematic analysis).**

| Thematic Category | n | Number of references |
|---|---|---|
| Category 1: Implementation of listening phases | 12 | ACCN 72 |
| Pre-listening | 4 | 17 |
| While listening | 8 | 35 |
| Post-listening | 3 | 20 |
| Category 2: The nature of listening material | 12 | ACCN 58 |
| Authentic | 3 | 9 |
| Non-authentic | 9 | 45 |
| Category 3: Characteristics of listening topics | 12 | ACCN 51 |
| Familiar and interesting | 4 | 13 |
| Not familiar and interesting | 8 | 38 |
| ACCN across the observation | 12 | 181 |

Note: n = the sample size of interview participants; ACCN = Aggregate Codes from Children Nodes.

**Table 10. Findings of close-ended observation (PLATO observation scales).**

| No. | Variables | Rating Scales | | | |
|-----|-----------|------|-----|------|----|
| | **Pre-listening** | **NEV** | **REV** | **SWEV** | **EV** |
| 1 | | 2 | 6 | 3 | 1 |
| 2 | | 1 | 6 | 4 | 1 |
| 3 | | 3 | 5 | 3 | 1 |
| 4 | | 4 | 3 | 4 | 1 |
| 5 | | 4 | 4 | 2 | 2 |
| | Total | 14 | 24 | 16 | 7 |
| | While-listening | NEV | REV | SWEV | EV |
| 6 | | 6 | 2 | 2 | 2 |
| 7 | | 1 | 2 | 4 | 5 |
| 8 | | 2 | 3 | 2 | 5 |
| 9 | | 3 | 3 | 5 | 1 |
| 10 | | 1 | 2 | 6 | 3 |
| 11 | | 2 | 2 | 4 | 4 |
| 12 | | 1 | 3 | 3 | 5 |
| 13 | | 1 | 2 | 5 | 2 |
| 14 | | 1 | 3 | 4 | 4 |
| 15 | | 4 | 3 | 6 | 1 |
| | Total | 22 | 25 | 41 | 30 |
| | Post-listening | NEV | REV | SWEV | EV |
| 16 | | 3 | 5 | 2 | 3 |
| 17 | | 4 | 5 | 3 | 0 |
| 18 | | 4 | 4 | 3 | 1 |
| 19 | | 5 | 4 | 3 | 0 |
| | Total | 16 | 18 | 11 | 4 |
| | Input for TLIS | NEV | REV | SWEV | EV |
| 20 | | 4 | 5 | 3 | 0 |
| 21 | | 0 | 1 | 6 | 5 |
| 22 | | 3 | 8 | 1 | 0 |
| 23 | | 3 | 7 | 1 | 1 |
| 24 | | 8 | 2 | 0 | 0 |
| Total | | 18 | 23 | 11 | 6 |

Note: NEV = Not Evident; REV = Rarely Evident; SWEV = Somewhat Evident; EV = Evident; TLIS indicates teaching listening integrated with speaking.

The participants noted that "listening is not commonly integrated with speaking in our class because the current format doesn't allow much practice with speaking. Students occasionally listen to read-aloud or use some audios, but explicit integration is rare" (T1, T9, T10, T11, and T12). Integration is typically planned, but not effectively implemented. A few participants (3) reported minimal integration, with T2 noting, "I record my voice and share it with students, but I'm not effectively using it for integrating listening and speaking." T4 and T6 mentioned attempting to use communicative activities to teach these skills.

Category 2 in "Table 11" reveals that most participants (n = 8) do not adapt listening materials for integrated listening and speaking classes. Instead, they primarily use a teacher-centered approach, such as engaging students in reading texts as quoted:

Table 11. Findings of interview data (thematic analysis in NVIVO).

| Thematic Category | n | Number of references |
|---|---|---|
| Category 1: Implementation of integrated listening and speaking | 12 | ACCN 69 |
| Yes | 3 | 18 |
| No | 9 | 46 |
| Category 2: Listening materials for integration | 12 | ACCN 79 |
| Authentic (adapted) | 4 | 19 |
| Non-authentic (not adapted) | 8 | 60 |
| Category 3: Strategy to motivate learners in integration | 12 | ACCN 60 |
| identifying learning styles | 2/10 | 8/23 (31) |
| techniques or activities to motivate students | 5/7 | 12/17 (29) |

Note: n indicates the sample size of interview participants, ACCN indicates aggregate codes from children's nodes.

> We never think about preparing material because of time limitations. We focus only on the module and are heavily reliant on previously designed material or existing curriculum. Unfortunately, some of the listening material does not have an audio version and must be read to students.

This concern was mentioned 60 times in the interviews, according to NVIVO analysis. It suggests that students have limited opportunities to engage actively in the classroom.

Conversely, a few participants adapted listening materials for integrated classes, using audio and video like news stories (T3) and various tasks from their computers (T4, T8, T9). Their opinions were mentioned 19 times, fewer than the 60 mentions in the second sub-theme.

Category 3 in "Table 11" highlights those strategies for motivating students in integrated aural-oral skills, focusing on learning styles and techniques. According to the data in the table, most participants (n = 10) teach integrated listening and speaking without identifying learning styles, noting that it requires more tasks and planning (T2, T3, T5, T6, and T11). Others articulated the belief that identifying needs is applicable, but lacks experience (T1, T7, T8, T9, and T12). These opinions were mentioned 23 times in the interview data.

However, a few participants (T4 and T10) noted that they rarely identify learners by their learning styles. T4 mentioned, "Sometimes I identify students' learning styles and adjust my approach beyond just reading the handout." T10 added, "I rarely do this, assuming auditory learners listen and visual learners watch videos to learn language skills." Only 8 occurrences were noted.

Regarding the techniques used by participants, T5, T6, T10, T11, and T12 shared their views 12 times in the data. For example:

> I try to make activities realistic, and sometimes I come up with related activities for listening and speaking skills integration. I use activities like dialogue, role-play, repetitive drills, filling gaps, dictation, and note-taking are helpful to integrate listening and speaking skills in the language classes.

Nevertheless, 7 participants did not mention developing tasks that effectively promote integrated listening and speaking activities. Their ideas were mentioned 17 times across the interviews. They reported and complained that:

> The tasks are not well-designed and not relevant. There is no balance between EFL listening and speaking tasks in English lessons. Since the purpose is to make the students talk in the classroom, EFL listening and speaking activities should be balanced, as these skills are interconnected.

## Discussions

The current study investigated University English as a Foreign Language Instructors' Implementations in Teaching Integral Listening with Speaking at Arba Minch and Wachemo Universities. A study indicated a notable incongruence between EFL instructors' understanding of the importance of integral teaching of EFL listening and speaking and their reported practices. Likewise, students' data illustrated different results from the teachers' survey results. This is aligned with a study where language teachers often accompany mismatched implementation of integral listening and speaking teaching practices, unlike their beliefs [112].

There was a mismatch between students' and teachers' data. This could indicate students' misperception and attitudes to learn the language communicatively or in real-life using listening input for speech production. This seems to be a study on students' perceptions and attitudes towards CLT learning [25].

To begin with, predictor selection for detailed discussion, the selection process depended on the previous literature as a theoretical base. The literature revealed that the predictor selection process should focus on forward and backward selections. Similarly, it should optimize and reduce overfitting in regression analysis models dealing with multicollinearity issues (ridge technique), and should shrink the structured and unstructured coefficients to balance complexity in regression models (Lasso technique) [100–102]. These methods or techniques align with the present study's process of predictor selection. Consequently, predictors such as features of listening materials, implementing listening phases, and using language features were identified and discussed below.

The present study highlighted that few EFL teachers implement regular engagement with varied audiovisual materials, i.e., listening features (one of the predictors integrated listening and speaking), where learners are more satisfied and motivated by the diverse listening topics [113]. A study concluded that the practice of student-centered activities used in listening increased actual practice and positively impacted the teaching and learning of English [20]. Conversely, the current study's findings revealed the reverse, except that teachers have positive perceptions of the frequent integral practices of teaching listening and speaking. Like the qualitative data, EFL students' results showed inconsistent findings with quantitative results obtained from teachers. This means that students' data supported the minimal practice of authentic listening materials. Instructors typically use reading texts and rely heavily on pre-determined activities and existing materials. Studies supported this finding that EFL teachers often do not incorporate authentic materials like technology integration to increase learners' learning autonomy in their classes [113]. Most participants focused on content rather than language; the class environment was dominated by EFL teachers and lecture techniques, reflecting a lack of motivation to encourage practical language use. Thus, learners showed minimal active participation [58,113]. This implies that the challenges of this mismatch were that, except for a few teachers, most language teachers in the university were technology resistant, focusing on the existing material. As a result, learners face challenges with listening proficiency and oral production.

The findings illustrated the importance of audio-visual learning for integrating listening and speaking in EFL classrooms. However, the implementation was unsuccessful despite EFL teachers' positive attitudes towards integrated aural-oral skills [28]. The use of activities based on authentic technology tools was ineffective in English courses, such as "Communicative English Language Skills I" in the present study [114]. Similarly, a study indicated it was not evident when language teachers used language learning apps for task-based authentic purposes [115].

Studies emphasize that familiar and engaging topics enhance integral listening and speaking in real-life contexts or authentic environments [67,116]. Familiar content improves L2 or FL fluency, unlike unfamiliar content hinders learning [67]. However, our findings showed that instructors rarely use such topics, which is inconsistent and aligned with a study that indicated that effective utilization of audio materials in aural-oral classes was not observed according to teachers' beliefs [117]. Learners use familiar language and content to transition into new subjects and challenges, aiding their adaptation to language learning. It is pointed out that starting with familiar input and progressing to new content enhances linguistic and cognitive abilities [65]. Similarly, in "Adapting Texts in CLIL Materials," this approach was not evident in the present study [35]. EFL teaching incorporating video could develop active learning and be effective for the EFL context.

Still, this study used only survey data, which might be rich enough to determine and show implications [118]. However, our mixed findings indicated that practical implementation was mismatched with teachers' belief systems, though authentic listening material significantly improves EFL learning in an integrated skill approach.

A few studies emphasized the efficiency of implementing three stages (predict integral listening, and speaking) [59]. In the present study, EFL teachers spend the most time in the while-listening phase, a study noted similarly [50]. Nevertheless, explicit instruction of pre-, while, and post-listening phases enhances language learning, like integrated listening and speaking [36]. However, the present study found inconsistent results, with pre- and post-listening phases rarely practiced. Pre-listening activities, which activate schemata and improve listening comprehension, were infrequently used, contradicting one of the findings [50]. Students' results in the present study confirmed that teachers rarely focused on effectively implementing listening stages to facilitate skills integration. A study supported this finding, noting the minimal opportunity for conscious implementation of the pre-listening phase [50].

EFL instructors with positive beliefs effectively use listening phases and activities, leading to meaningful learning and successful skills integration [119]. It is similar to bottom-up and top-down approaches that increase learning and listening processes [10]. However, the present study found inconsistencies where pre- and post-listening phases were rarely evident, with instructors not often using brainstorming tasks before lessons or assessing comprehension orally after listening. Studies signified that classroom practice habitually lacks focus on integrating listening with speaking through these phases [26]. This discrepancy aligns with the previous study findings [27], emphasizing the gap between perceived and actual practices. Like these studies, our study indicates that EFL instructors generally practice listening phases, but show considerable variability. This indicates the problem or misperception with the explicit integral listening and speaking instruction in the classroom. This could require using a model like the 3P model (presage, process, and product) for language instruction, which could boost holistic language teaching [120]. Similarly, this model can entertain blending learning, which enhances learning the language and the language features, particularly listening and speaking integrally and independently [121], the mismatch observed with the present study, but it focused on the effectiveness of the authentic listening.

The findings of the present study posited that while knowledge elements like linguistic, discourse, pragmatic, etc., these features were one of the predictors in the present study, can enhance the teaching and learning of listening and speaking, practical application aligned with these beliefs was rarely observed. Parallel research revealed that EFL university students were generally deficient in discourse features, impacting classroom interactions [122]. Studies showed difficulties in applying discourse knowledge [42]. However, a few studies showed a positive perception of discourse knowledge and moderate practical implementation, contrasting with the results of this study [123]. Moreover, our study did not focus on discourse elements like pragmatic and linguistic knowledge, despite instructors holding positive beliefs. Like our findings, studies depicted clear instructional practice using pragmatic knowledge is infrequent [45,46], unlike studies that illuminated the importance of pragmatic knowledge for improving aural-oral fluency in EFL classes [124]. Although linguistic ability (grammar and vocabulary) is crucial, teachers and students displayed significant variability in their responses. Teachers indicated they rarely integrate grammatical and vocabulary features into their teaching of listening and speaking, and learners' responses aligned with this view [125].

Quantitative reports from EFL teachers reveal that they rarely use task-based instruction (TBI) and content-based instruction (CBI) for integrated listening and speaking. This is parallel to the students' quantitative findings. Despite instructors' positive beliefs in these models, their application remains infrequent, differing from previous studies [61]. While both TBI and CBI can enhance listening and build speaking confidence [126], the study showed that TBI improved spoken fluency compared to the current English fluency of learners [127]. Although EFL instructors reported their regular use of task-based instruction (TBI), they did not employ integrated tasks in all listening phases. While TBI supports integrated skills instruction, studies show inconsistent findings with the present study, although TBI is recognized as crucial for enhancing aural-oral interaction [127].

Studies on the vlog portfolio project showed that students used scaffolding techniques to support peers and knowledgeable learners. It was found that learners' desires were met [70,128]. However, similar to our study, gaps in scaffolding techniques for language learning were observed, though EFL instructors positively believed in incorporating this technique. This study was a case study on the scaffolding technique that disclosed an impact on EFL learners' speaking skills. Teachers with positive perceptions were likely to implement this technique, agreeing on its benefits for enhancing speaking skills [129]. Nevertheless, the present study revealed significant variability and higher dispersion in responses, particularly from students.

## Conclusions and recommendations

Our study aimed at "Investigating University English as a Foreign Language Instructors' Implementations in Teaching of Integral Listening with Speaking" at Arba Minch and Wachemo Universities in Ethiopia. It involved a mixed-methods research approach for a comprehensive understanding. The findings identified: (a)A notable incongruence between EFL teachers' responses about integral teaching practices of listening and speaking and their implementation; (b) EFL learners reported concerning infrequent classroom practices while teachers recognized regular practices (implementation) of integral listening and speaking (B = 35.1%, Beta = 0.316); (c) using real-life listening materials and interesting listening topics impact integral listening and speaking instruction at B = 30.8%, Beta = 0.250; (d) Infrequent and ineffective implementation of listening phases (pre- and post-listening) at B = 20.4%, Beta = 0.203; and (e) Influencing factors that potentially hindered actual integral teaching of listening and speaking were identified including lack of resources, inadequate material adaptation, and limited student motivation.

Moreover, the ANOVA result revealed that the eta-squared = 0.750 and Cohen's d = 1.732. These results showed higher effect sizes. The value of era-squared (0.750) suggests that the teaching practice had a substantial influence on integral listening and speaking skills, accounting for 75% of the change ($\eta^2$ = 0.750), showcasing a very high effect, or there is a strong relationship between teaching practice and integral listening and speaking. This value exceeds conventional benchmarks of [130], where (small effect = 0.01, medium effect = 0.06, and large effect = 0.14). Similarly, Cohen's d = 1.732, which is a large effect size. This suggests substantial differences between the two groups (teacher and student). Regarding Cohen's d standard values, [130] states that small effect = 0.2, medium effect = 0.5, and large effect = 0.8. The results of *d* and $\eta^2$ aligned with the qualitative findings, where the practice revealed an influential variable to integral listening and speaking in EFL classes.

To conclude, the authors argued that improving the learning of EFL skills, particularly listening and speaking, requires an integrated instructional approach. This underscores the need to effectively incorporate explanatory variables, such as utilizing language features, effective utilization of authentic listening materials, and implementing listening phases, which were theory and data-driven themes. However, findings showed that Arba Minch and Wachemo Universities' EFL teachers paid little attention to understanding the roles of these explanatory variables that develop the integral teaching practices of EFL listening and speaking. Except for agreeing (were believed) on the importance of integral listening and speaking for effective language pedagogy, EFL instructors in the present study settings did not reflect in the application of English classes.

### Recommendations for further research

After investigating University English as a foreign language instructors' implementations in teaching integral listening and speaking at Arba Minch and Wachemo Universities, the present study identified the following areas that need to be researched further.

• Though the study was comprehensive, incorporating mixed methods approaches, its results suggest that further study should employ comparative tests, like t-tests, to scrutinize the group differences.

- Future research should utilize statistical validation through pre- and post-tests, longitudinal, and comparative studies to track improvements in learning listening and speaking skills together over time. This could help examine the relationship between variables, i.e., implementation of integral listening and speaking (IV) and explanatory variables indicated in this study, such as language features, effective utilization of authentic listening materials, and effective implementation of listening phases.

- The study suggests that future studies employ rubric-based speaking and listening assessments (additional data sources) to incorporate multiple observers. This could minimize or avoid a bias and could increase the validity and reliability of findings.

- Future studies should investigate a mismatch between instructors' self-reports and students' feedback of quantitative data results, and the qualitative data findings, particularly observations, which the present study did not cover. This could require a longitudinal study.

- It is recommended that the present research work be extended with a longitudinal research design. This would provide a richer dataset incorporating variables, particularly integral to listening and speaking, to explore the implementation of classroom pedagogy. To develop the explanatory power $R^2$, the study should employ an alternative model like mixed-effects models, which was highlighted in the limitations and implications sections.

- Even though eta-squared or Cohen's d is a potential measure of effect size and helps boost the understanding of the magnitude and significance of the findings, it might have a bias, such as a positive bias. To minimize bias concerning $\eta^2$ or Cohen's d, future studies should employ alternative measurements such as Epsilon and Omega squares.

- Further study should incorporate more or as many public universities as possible in Ethiopia. This could improve the generalizability of findings to other contexts.

- University EFL teachers should minimize the overreliance on non-authentic listening materials. This is because non-authentic materials might discourage learners' engagement and motivation to learn listening and speaking together in the classroom.

- Curriculum developers should explicitly consider the effectiveness of integral listening and speaking and incorporate them into the teaching syllabus. This could provide practical insights into the teaching strategy for EFL instructors.

- EFL teaching and learning programs should focus on training university instructors to utilize explanatory variables outlined in the study, such as language features, effective utilization of authentic listening materials, and implementation of listening phases to develop integral listening and speaking. This might advance language pedagogy and learning outcomes.

- EFL teachers should be responsible for being aware of learners' roles in teaching listening with speaking integrally, which improves learning outcomes, in addition to learners' responsibility to learn the language employing integral listening and speaking.

- Stakeholders at Arba Minch and Wachemo Universities should provide an environment that facilitates the effective utilization of learning resources that promote real-life learning of English in a spoken context.

  - The present researchers should share the findings of this study with a larger audience beyond the current study contexts, like other public universities in Ethiopia and the international community.

## Limitations and implications

**Limitations.** A few limitations hindered the present study despite the inspiring findings. The statistical analysis is rigorous and appropriate, involving a mixed-methods approach (used mean and standard deviation, and multiple

regression analysis for surveys. This could help examine the relationship between explanatory and outcome variables. However, the study has limitations in that no comparative tests, like t-tests, were used to scrutinize the group differences. The study did not employ statistical validation to see the improvement of learning outcomes in integral listening and speaking. Thus, it would have been better if our research employed pre- and post-tests, longitudinal, and comparative studies to track improvements in students' listening/speaking skills over time. Likewise, employing rubric-based evaluations also gives the researcher another layer of evidence because it provides systematic, structured decisions from EFL experts. It can also help cross-check whether the observed or self-reported progress matches actual performance. It would have been better if the study used rubric-based assessment as an additional data source for validation for assessing integral listening and speaking. This could incorporate multiple observers, unlike the present research, which employed a single observer, and could be subject to bias.

There is a mismatch between instructors' self-reports (responses) and students' feedback. For instance, instructors report a higher practice of integral listening and speaking skills, but students' survey responses and classroom observations contradict this. This could suggest bias in self-reported data from the two groups (instructor and student). Likewise, the quantitative data reported a higher instructor self-rating (mean values), but qualitative findings illustrated poor implementation of integral listening and speaking. These issues could affect the findings and conclusions of the study.

Self-reported data in this study could lead to biased findings of adjusted $R^2$. This shows the confounding factors due to self-reported data, which might affect the value of $R^2$. Similarly, sometimes, a lower $R^2$ value might highlight the inadequacy of the model utilized, i.e., it could be measurement error. In the present study, the association between dependent and independent variables might not be firmly linear, and future research could explore the integral listening and speaking about their real application in EFL classrooms. This scenario calls for alternative models to minimize, if possible, to avoid the measurement complexity and improve the value of adjusted $R^2$, like leave-one-out $R^2$ (LOO $R^2$) or mixed-effect models.

Moreover, beyond reporting the p-value in a regression analysis, reporting the effect sizes (Cohen's d, eta squared) is important to increase the understanding of the magnitude and actual significance of the finding [131]. However, it could have been better if the study utilized the Epsilon and Omega squares for further enrichment to improve the magnitude of the prediction of variables. Therefore, the study acknowledges this as one of the limitations.

Our study focuses on or is geographically limited to only two Ethiopian universities, i.e., Arba Minch and Wachemo Universities. This could make its broader applicability uncertain and constrain the generalizability of its findings.

The study reported the overreliance on non-authentic listening materials. This might discourage learners' engagement and motivation to learn listening and speaking together. Further, the qualitative findings revealed that there were limited aural resources, i.e., though there were some resources like the language laboratory, most were not functional, except for installation. This could hinder meaningful language learning in a spoken context in the classroom.

**Implications.** The present study aimed at "Investigating University English as a Foreign Language Instructors' Implementations in Teaching Integral Listening with Speaking" at Arba Minch and Wachemo Universities in Ethiopia. The study points out key theoretical and pedagogical implications for EFL teachers at Arba Minch and Wachemo Universities in Ethiopia and a broader context beyond the current settings.

To begin with, theoretical implications, our study will help as a foundational step for further studies, emphasizing integral listening and speaking. It encourages mixed-methods researchers because it offers valid instruments such as questionnaires, interview guides, and observation checklists written English. This could save time and minimize the fatigue of other researchers in this area. The present study will disclose evidence in the current literature (comparing previous studies with the present findings), in line with language skills integration (listening and speaking focused) to increase language learning (this contributes to the body of knowledge in the field of study).

Further, this research provides valuable insights into EFL teaching/learning challenges in Ethiopian universities, curriculum developers, and policy makers. For example, findings indicate that teaching/learning integral listening and speaking can be boosted if teachers and students effectively utilize language features, authentic listening materials, and listening phases (explanatory variables). Thus, teachers can encourage learners to learn the language by providing authentic

listening tasks that enhance skills integration (real-life learning) in the classroom. This requires adapting listening materials for integrated skills instruction rather than relying solely on the existing teaching module. Hence, the study underscores the need for authentic resources and professional development that highlight integral listening and speaking. This could minimize the disparities between teaching and learning implementation of integral listening and speaking.

Although the model's explanatory power in our study is weak, it could help identify the statistically significant relationship between integral listening and speaking (dependent variable) and its implementation in the classroom (independent variable) rather than to predict learning outcomes thoroughly. This indicates meaningful insight, even though the overall change explained was moderate (28.8%). Further, unobservable factors due to self-reported data could affect the value of adjusted $R^2$. This might help indicate the limitation and suggest further study to fill this gap, employing eta-squared. The present study recognizes the practical significance of using eta-squared, which could boost the explanatory power of the variables as it helps remove or minimize the bias of the effect size of language and psychological research [132,133].

Curriculum reviewers might benefit from our study's findings, which could help them incorporate explanatory variables mentioned in this study that could advance language teaching and learning (aural-oral proficiency).

## Supporting information

**S1 Appendix. Instructors' questionnaire.**
(DOCX)

**S2 Appendix. interview.**
(DOCX)

**S3 Appendix. Students' questionnaire.**
(DOCX)

**S4 Appendix. Observation checklist.**
(DOCX)

**S5 Appendix. The three-cycle coding methods are summarized.**
(DOCX)

**S6 Appendix. Interview transcription.**
(DOCX)

## Acknowledgments

The authors thank all English Language and Literature Department instructors of Arba Minch and Wachemo Universities for their willingness and participation in the present study.

The authors' membership is listed on the title page: Addisu Bogale Shago, Elias Woemego Bushisso, and Taye Gebremariam Olamo. The corresponding author (PhD Candidate) thanks the co-authors for their guidance in writing this manuscript, mentioning their roles on the title page.

## Author contributions

**Conceptualization:** Addisu Bogale Shago.

**Data curation:** Addisu Bogale Shago.

**Formal analysis:** Addisu Bogale Shago.

**Investigation:** Addisu Bogale Shago.

**Methodology:** Addisu Bogale Shago.

**Software:** Addisu Bogale Shago.

**Supervision:** Elias Woemego Bushisso, Taye Gebremariam Olamo.

**Validation:** Elias Woemego Bushisso, Taye Gebremariam Olamo.

**Writing – original draft:** Addisu Bogale Shago.

**Writing – review & editing:** Elias Woemego Bushisso, Taye Gebremariam Olamo.

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
