## [Decision Letter · Decision Letter 0]

27 Feb 2025

PONE-D-24-53961Investigating University English as a Foreign Language Instructor's Implementations in Teaching Integral Listening with SpeakingPLOS ONE

Dear Dr. Shago,

Thank you for submitting your manuscript to PLOS ONE. After careful consideration, we feel that it has merit but does not fully meet PLOS ONE’s publication criteria as it currently stands. Therefore, we invite you to submit a revised version of the manuscript that addresses the points raised during the review process.

We look forward to receiving your revised manuscript.

Kind regards,

Dawit Dibekulu, PhD

Academic Editor

PLOS ONE

Journal Requirements:

“We also appreciate Hawassa University and Wolaita Sodo University for financial support for data collection. “

“The author(s) received no specific funding for this work”

5. Please ensure that you include a title page within your main document. You should list all authors and all affiliations as per our author instructions and clearly indicate the corresponding author.

6. Please amend your manuscript to include your abstract after the title page.

7. Please include a copy of Table 9 which you refer to in your text on page 22.

8. Please include captions for your Supporting Information files at the end of your manuscript, and update any in-text citations to match accordingly. Please see our Supporting Information guidelines for more information: http://journals.plos.org/plosone/s/supporting-information .

Reviewers' comments:

Reviewer's Responses to Questions

**Comments to the Author**

1. Is the manuscript technically sound, and do the data support the conclusions?

Reviewer #1: Partly

Reviewer #2: Partly

2. Has the statistical analysis been performed appropriately and rigorously? 

Reviewer #1: Yes

Reviewer #2: No

3. Have the authors made all data underlying the findings in their manuscript fully available?

Reviewer #1: No

Reviewer #2: Yes

4. Is the manuscript presented in an intelligible fashion and written in standard English?

Reviewer #1: No

Reviewer #2: Yes

5. Review Comments to the Author

Reviewer #1: Dear Authors,

Thank you for your submission. Below are my detailed comments regarding the manuscript:

1. Technical Soundness & Data Support for Conclusions (Partly)

The manuscript presents a valuable study on integrating listening and speaking skills in Ethiopian EFL classrooms. The mixed-methods approach (surveys, interviews, classroom observations) provides insightful findings, but several limitations weaken the strength of the conclusions:

Mismatch Between Instructor Perceptions and Student Feedback:

Instructors report high integration of listening and speaking skills, but students' survey responses and classroom observations contradict this.

This suggests possible overestimation or bias in self-reported data.

Recommendation: The study should explicitly acknowledge this discrepancy and provide further validation, such as external assessments.

Lack of Longitudinal Data or Experimental Control:

The study does not track improvements in students’ listening/speaking skills over time.

No control group is used to compare different instructional strategies.

Recommendation: Future research should consider pre/post-assessments or comparative studies to strengthen causal claims.

Limited Generalizability:

The study focuses on only two Ethiopian universities, making broader applicability uncertain.

Recommendation: The authors should discuss this limitation and suggest cross-institutional studies for future work.

2. Statistical Analysis (Yes)

The statistical analysis is rigorous and appropriate, employing:

Descriptive statistics (means, standard deviations) for survey responses.

Multiple regression analysis to examine the relationship between instructional practices and skill integration.

Normality checks and collinearity diagnostics (VIF & Tolerance).

However, areas for improvement include:

No comparative tests (e.g., t-tests, ANOVA) to examine group differences.

No statistical validation of actual student improvement (e.g., pre/post-tests).

Recommendation: Including comparative analysis and pre/post performance data would enhance empirical validation.

3. Data Availability (No)

The full dataset (survey responses, coded qualitative data, regression outputs) is not publicly available.

Interview transcripts and NVIVO-coded themes are not included.

Recommendation:

The authors should deposit anonymized raw data in a public repository (e.g., Figshare, Zenodo, Dryad).

If privacy concerns exist, structured data summaries should be provided as supplementary materials.

4. Language and Clarity (No)

The manuscript contains multiple grammatical errors, awkward phrasing, and redundant sections.

Overly complex sentence structures make comprehension difficult.

Terminology is inconsistent (e.g., "integral teaching" vs. "integrated teaching").

Recommendation:

The manuscript should undergo thorough professional copyediting to improve clarity and readability.

Redundant content should be streamlined for better flow.

Technical terms should be consistently defined and explained.

5. Additional Concerns and Recommendations

Ethical Considerations: The study states that ethical approval was obtained, but more clarity on informed consent procedures (especially for student participants) would be beneficial.

Study Contribution: The research provides valuable insights into EFL teaching challenges in Ethiopian universities, making it relevant to curriculum developers, instructors, and policymakers.

Recommendation for Strengthening Findings:

Explicitly acknowledge biases in self-reported data.

Use additional data sources for validation (e.g., rubric-based speaking assessments).

Include more structured comparisons with prior research.

Final Recommendation: Major Revision

The study is well-conceptualized and methodologically sound, but significant revisions are required before acceptance. Key areas to address:

Data availability compliance with PLOS ONE policy.

Improvements in empirical validation (pre/post assessment, additional statistical tests).

Grammar, clarity, and structural issues requiring professional copyediting.

Reviewer #2: Manuscript Title:

Investigating University English as a Foreign Language Instructor’s Implementations in Teaching Integral Listening with Speaking

1. Overall Assessment

This manuscript examines how university EFL instructors implement integral teaching of listening and speaking, a crucial area in applied linguistics. However, the paper contains significant methodological flaws, imprecise language, inadequate engagement with recent literature, and problematic statistical analyses. There are also major inconsistencies between the research objectives, findings, and conclusions. Without substantial revisions, the manuscript does not meet the standard for publication.

2. Major Language and Structural Issues (By Section & Page Number)

Abstract (Page 1)

•The abstract is wordy and lacks precision. The phrase “scarce studies in the area” should be replaced with “limited prior research.”

•The sentence “Students’ data confirmed these findings” lacks clarity. Which specific data points confirm the findings? Be explicit.

•Typos:

•“forwarded recommendations” → should be “provided recommendations.”

•“target variables that improve integral instruction” → should be “key factors influencing integrated instruction.”

Introduction (Pages 2-4)

•The introduction lacks a clear problem statement. It vaguely suggests that integrated teaching is underutilized without presenting quantifiable evidence or literature-based justification.

•Inconsistent terminology: The manuscript inconsistently switches between “integral teaching” and “integrated teaching.” Choose one term and maintain consistency.

•Overuse of generic phrases:

•“Speaking and listening are significant skills for learning other English language skills” → This is redundant; revise to “Speaking and listening are foundational for English language acquisition.”

•Suggested References to Incorporate:

•Al Fraidan (2025). Evaluating Lexical Competency in Saudi Arabia’s Hybridized EFL Ecosystem: A Taxonomic Exploration of Vocabulary Assessment Modalities and the Imperative for AI-Enhanced Adaptive Testing.

•Why? This study provides insights into language skill integration in diverse learning environments, directly relevant to the manuscript’s topic.

Methodology (Pages 5-7)

•Sampling Issues:

•The study claims to have used “systematic sampling” for student participants but does not explain the criteria for inclusion/exclusion.

•The total sample size of 252 participants is mentioned, but how were these numbers determined? Was a power analysis conducted?

•Data Collection Methods:

•The manuscript states that “observations were structured”, but there is no mention of an observation rubric or inter-rater reliability measures.

•NVIVO was used for qualitative analysis, but the coding framework is not provided. What themes were pre-identified? Were codes data-driven or theory-driven?

•Statistical Problems:

•Regression models lack justification for predictor selection.

•VIF values are reported but not interpreted correctly. The manuscript does not discuss potential collinearity issues in explanatory variables.

•Suggested References to Incorporate:

•Al Fraidan & Alsubaie (2025). Exam Anxiety and Vocabulary Challenges: Insights from Postgraduate Female Students in Open and Closed Book Exams.

•Why? The study provides empirical insights into student learning challenges, essential for contextualizing the findings.

Findings and Discussion (Pages 10-14)

•Mismatch Between Quantitative and Qualitative Findings:

•Table 3 reports high instructor self-ratings (mean values around 3.7-4.0), yet qualitative findings suggest poor implementation.

•There is no attempt to reconcile this contradiction—why do students and teachers perceive instructional practices so differently?

•Statistical Errors:

•The adjusted R² value of 0.288 in the regression model suggests that only 28.8% of the variance is explained. This is weak. What other variables might contribute to implementation differences?

•No effect sizes are reported. Effect sizes (Cohen’s d, eta squared) are critical in educational research.

•Discussion Fails to Engage with Prior Studies:

•There is no meaningful comparison with existing research. The discussion simply repeats the results without analyzing how they align with or challenge previous work.

•Suggested References to Incorporate:

•Al Fraidan (2024). The impact of students’ misperceptions on test performance: A 3P model and self-determination theory approach.

•Why? This study discusses perception discrepancies in learning contexts, relevant for explaining the conflicting quantitative and qualitative results in the manuscript.

Limitations and Recommendations (Pages 15-16)

•Limitations Section is Underdeveloped:

•The manuscript vaguely states “instructors were theoretically grounded, but not reflected in practical application.” This needs quantifiable evidence.

•The study’s geographical limitation to two Ethiopian universities is not adequately acknowledged as a constraint to generalizability.

•Recommendations are Not Practical:

•The study suggests curriculum reform but fails to outline specific pedagogical changes. How should universities modify teaching strategies?

•Suggested References to Incorporate:

•Al Fraidan & Al-Harazi (2023). Using Social Media Platforms to Prepare for Examinations Post-COVID-19: The Case of Saudi University EFL Learners.

•Why? This research presents alternative pedagogical tools, which could strengthen the manuscript’s recommendations.

3. Specific Recommendations for Major Revisions

1.Improve clarity and precision in writing. Revise abstract and introduction to eliminate redundancy.

2.Clearly define key terms. Provide a consistent definition of “integral teaching” and maintain terminology throughout.

3.Enhance methodological transparency.

•Justify sample size selection and recruitment criteria.

•Explain NVIVO coding procedures for qualitative analysis.

•Provide inter-rater reliability data for observational measures.

4.Strengthen statistical analysis.

•Report effect sizes and confidence intervals.

•Discuss collinearity concerns in regression models.

•Address the weak explanatory power (R² = 0.288) in regression analysis.

5.Improve discussion section.

•Compare findings with existing research rather than merely repeating results.

•Explain discrepancies between instructor self-reports and student evaluations.

6.Provide concrete recommendations. Suggest practical, actionable steps for improving teaching practices.

7.Incorporate recent literature. Include the suggested references above

4. Final Decision: Major Revisions Required

The manuscript fails to meet the standard for publication in its current form due to methodological flaws, statistical inconsistencies, and poor engagement with existing literature. Substantial revisions are necessary before reconsideration.

6. PLOS authors have the option to publish the peer review history of their article (what does this mean? ). If published, this will include your full peer review and any attached files.

**Do you want your identity to be public for this peer review?** For information about this choice, including consent withdrawal, please see our Privacy Policy .

Reviewer #1: **Yes: ** Mohanad Aljbour

Reviewer #2: No

---

## [Author Response · Author response to Decision Letter 1]

28 Apr 2025

Manuscript Revision (PONE-D-24-53961)

Response to Reviewer 1

Thank you so much for your valuable comments you provided for our manuscript to improve its quality for consideration for publication. In the following section.

Comment (general): Technical Soundness & Data Support for Conclusions (Partly)

The manuscript presents a valuable study on integrating listening and speaking skills in Ethiopian EFL classrooms. The mixed-methods approach (surveys, interviews, classroom observations) provides insightful findings, but several limitations weaken the strength of the conclusions:

Response: Yes, these limitations can affect the quality of the conclusion, thus, we seriously considered all your comments for betterment.

Comment 1: Mismatch between Instructor Perceptions and Student Feedback:

Instructors report high integration of listening and speaking skills, but students' survey responses and classroom observations contradict this.

This suggests possible overestimation or bias in self-reported data.

Recommendation: The study should explicitly acknowledge this discrepancy and provide further validation, such as external assessments.

Response: Thank you so much for your priceless insight. We accepted your comment and acknowledged it explicitly (see the discussion section, line numbers “829-832”, and p.39).

Comment 2: Lack of Longitudinal Data or Experimental Control:

The study does not track improvements in students’ listening/speaking skills over time.

No control group is used to compare different instructional strategies.

Recommendation: Future research should consider pre-/post-assessments or comparative studies to strengthen causal claims.

Response: We agree with your valuable comment. Thank you. We have recommended and included in limitations (see line numbers “941-942”, p.43).

Comment 3: Limited Generalizability:

The study focuses on only two Ethiopian universities, making broader applicability uncertain.

Recommendation: The authors should discuss this limitation and suggest cross-institutional studies for future work.

Response: Yes, exactly, we accepted it as is and incorporated limitations and recommended it for further research.

Comment 4: 2. Statistical Analysis (Yes)

The statistical analysis is rigorous and appropriate, employing:

Descriptive statistics (means, standard deviations) for survey responses.

Multiple regression analysis to examine the relationship between instructional practices and skill integration.

Normality checks and collinearity diagnostics (VIF & Tolerance).

However, areas for improvement include:

No comparative tests (e.g., t-tests, ANOVA) to examine group differences.

No statistical validation of actual student improvement (e.g., pre/post-tests).

Recommendation: Including comparative analysis and pre/post performance data would enhance empirical validation.

Response: Thank you so much for your comment in this regard. We have revised the manuscript considering your comment into account. The revision suggested and acknowledged the limitations (see line number “940”, p. 43).

Comment 5: 3. Data Availability (No)

The full dataset (survey responses, coded qualitative data, regression outputs) is not publicly available.

Interview transcripts and NVIVO-coded themes are not included.

Recommendation:

The authors should deposit anonymized raw data in a public repository (e.g., Figshare, Zenodo, Dryad).

If privacy concerns exist, structured data summaries should be provided as supplementary materials.

Response: Thank you for showing concern about the guideline, which is very important. Thus, we have uploaded these datasets

Comment 6: 5. Additional Concerns and Recommendations

Ethical Considerations: The study states that ethical approval was obtained, but more clarity on informed consent procedures (especially for student participants) would be beneficial.

Response: Thank you. We revised based on your comment (see line numbers “303-305”, p.13).

Comment 7: Study Contribution: The research provides valuable insights into EFL teaching challenges in Ethiopian universities, making it relevant to curriculum developers, instructors, and policymakers.

Recommendation for Strengthening Findings:

Explicitly acknowledge biases in self-reported data.

Use additional data sources for validation (e.g., rubric-based speaking assessments).

Include more structured comparisons with prior research.

Response: We appreciate your comment. This is significant for further study as it was not included in the first submission (see line numbers “937-939”, p.43). Limitations were acknowledged (see line numbers “981-987”, p.45).

Language issues, clarity, and structure were seriously considered in revision.

Manuscript Revision (PONE-D-24-53961)

Reviewer 2 Comments and Response to Comments

First of all authors are thankful and appreciate your precious time you exerted to read our manuscript to shape it. This would help improve our work, move one step ahead before considering it for publication. You provided priceless input for us to thoroughly revise the manuscript. Thank you so much. We tried our best accordingly in the following section.

Comment 1: Overall Assessment

This manuscript examines how university EFL instructors implement integral teaching of listening and speaking, a crucial area in applied linguistics. However, the paper contains significant methodological flaws, imprecise language, inadequate engagement with recent literature, and problematic statistical analyses. There are also major inconsistencies between the research objectives, findings, and conclusions. Without substantial revisions, the manuscript does not meet the standard for publication.

Response: Thank you so much for this insight. Yes, you are right because these issues should be considered for the quality of the manuscript, though we have missed. We understood the need for critical revision to enhance the quality of the manuscript. Accordingly, we made substantial revisions in each section based on your feedback. For example, in the methodology section, we make it rigorous to improve this section. Likewise, we thoroughly read it and edited the language for accuracy. In addition, we utilized Grammarly to enhance the quality; recent literature was incorporated, and so on.

Comment 2: Major Language and Structural Issues (By Section & Page Number)

Abstract (Page 1)

•The abstract is wordy and lacks precision. The phrase “scarce studies in the area” should be replaced with “limited prior research.”

•The sentence “Students’ data confirmed these findings” lacks clarity. Which specific data points confirm the findings? Be explicit.

Response: Thank you. We revised most sections to minimize wordiness. We replaced specific phrases as per your comment (see p.2, lines 28-29).

Comment 3: •Typos:

•“forwarded recommendations” → should be “provided recommendations.”

•“target variables that improve integral instruction” → should be “key factors influencing integrated instruction.”

Response: Thank you. Authors revised it accordingly, please see line numbers 40-41 and 45-46 on p.2. (Abstract section).

Introduction (Pages 2-4)

Comment 4: •The introduction lacks a clear problem statement. It vaguely suggests that integrated teaching is underutilized without presenting quantifiable evidence or literature-based justification.

Response: We appreciate the reviewer's valuable comment. Accepting this idea, we almost changed the introduction section. Particularly, stating or contextualizing the problem statement in line numbers ” 85-135”, in addition to being indicated in the literature section. Hopefully, it makes sense compared to our work before your comment. Your comment better shaped our introduction.

Comment 5: •Overuse of generic phrases:

•“Speaking and listening are significant skills for learning other English language skills.” → This is redundant; revise to “Speaking and listening are foundational for English language acquisition.”

Response: Yes, you are right. We corrected this comment, more sound than ours. Terminology should be consistent in our study (see line 71 on p.3). The suggested reference in the introduction section for help skills integration is incorporated (L96 and 100, [22] p4).

Methodology (Pages 5-7)

•Sampling Issues:

Comment 6: •The study claims to have used “systematic sampling” for student participants, but does not explain the criteria for inclusion/exclusion.

•The total sample size of 252 participants is mentioned, but how were these numbers determined? Was a power analysis conducted?

Response: Your comment sounds, and we agree. Thank you so much. Therefore, we have revised sample procedures to include sample size as we did not incorporate the edited copy and the later corrected version, which was our mistake. Considering your comment, we indicated revision from line numbers “358-445, 16…19.” Additionally, the total sample size of “252” is the combined total of teachers (n = 92) and students (n =160), and the determination process is indicated in these lines and pages. Power analysis was not conducted, and we have corrected it in the revision. Procedures were shown to determine the sample size.

Data Collection Methods:

Comment 7: •The manuscript states that “observations were structured”, but there is no mention of an observation rubric or inter-rater reliability measures.

Response: Thank you, dear reviewer, for this comment. In our study, only one observer (principal investigator, currently PhD candidate) was involved, thus, no inter-rater reliability was considered. Instead, we explained to use of PLATO observation scales as observation carried out by the first authors (PhD candidate) rather than observed by different observers or two more raters rate behaviors, but in this study, we have counted the frequency and numbered it, then interpreted the counted quantity based on the PLATO scale. Please see the revision on page 22, line numbers 504-513.

Comment 8: •NVIVO was used for qualitative analysis, but the coding framework is not provided. What themes were pre-identified? Were the codes data-driven or theory-driven?

Response: Thank you for an invaluable idea. We strongly agree with this comment and amended the manuscript to increase its quality, which is indicated on page 27, line numbers “618-637.”

Statistical Problems:

Comment 9: •Regression models lack justification for predictor selection.

Response: Yes, dear reviewer, it sounds nice. It would have been better if justified during submission, but we did consider it in this way. Thank you for this insight. Hence, we seriously amended our understanding of its importance. This idea was indicated in line numbers “593-604” on page 26. Accordingly, one theme was removed through the predictor section procedure. Thank you so much again.

Comment 10: •VIF values are reported, but not interpreted correctly. The manuscript does not discuss potential collinearity issues in explanatory variables.

Response: Yes, we agree that it was not interpreted. As per your insight, we made a change. Please see line numbers “491-497”. The suggested reference was also incorporated into the manuscript as it was significant in indicating empirical insights into student learning challenges, essential for contextualizing the findings.

Findings and Discussion (Pages 10-14)

•Mismatch between Quantitative and Qualitative Findings:

Comment 11: •Table 3 reports high instructor self-ratings (mean values around 3.7-4.0), yet qualitative findings suggest poor implementation.

Comment 12: •There is no attempt to reconcile this contradiction—why do students and teachers perceive instructional practices so differently?

Response: Thank you so much. We responded to these comments by merging. These differences should have been discussed during the first submission. Your comment helped us to revise, so we have incorporated in limitation and suggested further recommendations that current researchers have not covered, and included in the discussion section the reason for the mismatch, though it was not enough which initiate the researchers to show limitations.

•Statistical Errors:

Comment 13: •The adjusted R² value of 0.288 in the regression model suggests that only 28.8% of the variance is explained. This is weak. What other variables might contribute to implementation differences?

Response: Thank you, reviewer. This appeared weak, or in the case of lower R-square values, we can consider confidence intervals and VIF, which could balance this problem, though we haven’t used. VIF in our findings showed a good standard, but still needs a strong R-Squared value (line “494”, p.21). Thus, the limitation for these issues you have raised (see limitation section L955-956, p.44).

Comment 14: •No effect sizes are reported. Effect sizes (Cohen’s d, eta squared) are critical in educational research.

Response: We appreciate your comment. We understand that the Beta value seemed enough in the regression model. Like your concern, some studies urge its significance in indicating a magnitude influence. This appeared in our studies and suggested for further research (see line numbers “957-959”, p.44).

•Discussion Fails to Engage with Prior Studies:

Comment 15: •There is no meaningful comparison with existing research. The discussion simply repeats the results without analyzing how they align with or challenge previous work.

Response: Thank you. Based on your insight, we made changes in the discussion section. Indicated interpretations and cited current literature as suggested (see line numbers “873-878”, pp. 40-41). The implication challenge was referred to the mismatch (see line numbers “830-832”, p.39).

•Limitations Section is Underdeveloped:

Comment 16: •The manuscript vaguely states, “instructors were theoretically grounded, but not reflected in practical application.” This needs quantifiable evidence. •The study’s geographical limitation to two Ethiopian universities is not adequately acknowledged as a constraint to generalizability.

Response: Thank you. The first question was revised in the text of the manuscript, and the second was included in the limitation section as it should be done as you commented. (See line numbers “960-961”, p.44). Limitations was also included in this regard (see line numbers “1010-1012”, p.46).

• Recommendations are Not Practical:

Comment 17: •The study suggests curriculum reform, but fails to outline specific pedagogical changes. How should universities modify teaching strategies?

Response: Thank you so much. We have revised the recommendation section thoroughly because it was not comprehensive as you suggested.

NB: Specific Recommendations for Major Revisions indicated in 1-6 were included in the response (comments 1-17).

Comment 18: Incorporate recent literature. Include the suggested references above.

Response: All newly cited references were incorporated into the manuscript. Some not recent but cited for purpose of importance. Indicated as follows.

Newly incorporated References

Lee JS, Chen Hsieh J. University students’ perceptions of English as an International Language (EIL) in Taiwan and South Korea. Journal of Multilingual and Multicultural Development. 2018 Oct 21; 39(9):789-802.

Luczaj K, Leonowicz-Bukala I, Kurek-Ochmanska O. English as a lingua franca? The limits of everyday English-language communication in Polish academia. English for Specific Purposes. 2022 Apr 1; 66:3-16.

Al Fraidan A. Evaluating lexical competency in Saudi Arabia's hybridized EFL ecosystem: A taxonomic exploration of vocabulary assessment modalities. International Journal of Distance Education Technologies (IJDET). 2025 Jan 1; 23 (1):1-36.

Al Fraidan A, Fakhli I. Vocabulary learning strategies of university students: The case of preparatory year students and English major students. Porta Linguarum. 2024 January; 2024 (41): 45-63.

Al Fraidan A. The impact of students’ misperceptions on test performance: A 3p model and self-determination theory approach. Edelweiss Applied Science and Technology. 2024; 8(5):1773-84.

O’Connor C, Joffe H. Intercoder reliability in qualitative research: Debates and practical guidelines. International journal of qualitative methods. 2020; 19 (1-13): 1-13. Available from: https://doi.org/10.1177

---

## [Decision Letter · Decision Letter 1]

18 May 2025

PONE-D-24-53961R1Investigating University English as a Foreign Language Instructor's Implementations in Teaching Integral Listening with SpeakingPLOS ONE

Dear Dr. Shago,

Thank you for submitting your manuscript to PLOS ONE. After careful consideration, we feel that it has merit but does not fully meet PLOS ONE’s publication criteria as it currently stands. Therefore, we invite you to submit a revised version of the manuscript that addresses the points raised during the review process.

We look forward to receiving your revised manuscript.

Kind regards,

Dawit Dibekulu, PhD

Academic Editor

PLOS ONE

Journal Requirements:

Additional Editor Comments :

Dear Author(s),

Thank you for your submission. Please note that one reviewer comment has been added requesting evisions to improve clarity and strengthen the discussion. Kindly address this in your revised manuscript.

Best regards,

Reviewers' comments:

Reviewer's Responses to Questions

**Comments to the Author**

1. If the authors have adequately addressed your comments raised in a previous round of review and you feel that this manuscript is now acceptable for publication, you may indicate that here to bypass the “Comments to the Author” section, enter your conflict of interest statement in the “Confidential to Editor” section, and submit your "Accept" recommendation.

Reviewer #1: All comments have been addressed

Reviewer #2: All comments have been addressed

2. Is the manuscript technically sound, and do the data support the conclusions?

Reviewer #1: Yes

Reviewer #2: Yes

3. Has the statistical analysis been performed appropriately and rigorously? 

Reviewer #1: Yes

Reviewer #2: Yes

4. Have the authors made all data underlying the findings in their manuscript fully available?

Reviewer #1: Yes

Reviewer #2: Yes

5. Is the manuscript presented in an intelligible fashion and written in standard English?

Reviewer #1: No

Reviewer #2: Yes

6. Review Comments to the Author

Reviewer #1: There is a significant mismatch between instructors’ self-reports and students’ feedback, suggesting possible overestimation in instructor perceptions. While this discrepancy is acknowledged in the discussion, it would benefit from deeper triangulation with additional measures, such as rubric-based performance evaluations or third-party assessments.

The conclusions are generally consistent with the data, but should be framed more cautiously due to the descriptive and cross-sectional nature of the research design.

The regression model has limited explanatory power (Adjusted R² = 0.288), and this should be more critically discussed in terms of implications.

No inferential group comparisons (e.g., t-tests or ANOVA) were performed to examine differences across groups, which would have strengthened the analysis.

Effect sizes (Cohen’s d, eta squared) were not reported, limiting interpretation of practical significance.

The selection of regression predictors was not well justified initially; this was later addressed, but could still benefit from clearer theoretical alignment.

Despite the authors’ statement that "all relevant data are within the manuscript and supporting information files," the underlying data are not fully available:

No anonymized raw data from surveys (Likert responses) are included.

No SPSS or regression output files are provided.

The NVIVO-coded qualitative data or coding frameworks are not attached.

The data are not deposited in any public repository, which is preferred under PLOS ONE’s data policy.

The authors must upload all underlying data (survey responses, regression model variables, and qualitative coding schema) to a recognized open-access repository (e.g., Zenodo, Figshare) or provide them as supplementary files.

The revised version has improved in clarity and readability. However, a few sections—particularly in the methodology and discussion—would benefit from tighter organization and more concise phrasing. Please ensure final proofreading for grammatical consistency.

Recommendation

The manuscript presents meaningful findings and has potential for publication. However, it requires moderate revision to address the following:

Provide public access to underlying data in line with PLOS ONE’s policy.

Include or discuss effect sizes and comparative analyses to strengthen statistical interpretation.

Frame conclusions with appropriate caution, given the cross-sectional, descriptive nature of the study.

Reviewer #2: (No Response)

7. PLOS authors have the option to publish the peer review history of their article (what does this mean? ). If published, this will include your full peer review and any attached files.

**Do you want your identity to be public for this peer review?** For information about this choice, including consent withdrawal, please see our Privacy Policy .

Reviewer #1: **Yes: ** Mohanad Aljbour

Reviewer #2: No

---

## [Author Response · Author response to Decision Letter 2]

27 May 2025

Date: 26 May 2025

Journal name: PLOS ONE

Article type: Original Research Article

Reviewer #1

Subject: The Response to Reviewer 1 Comments

Dear Reviewer! My respectful greetings go to you!

I, the corresponding author of the manuscript entitled “Investigating University English as a Foreign Language Instructor's Implementations in Teaching Integral Listening with Speaking” (PONE-D-24-53961R1), would like to appreciate your constructive feedback and the valuable time you devoted to improving our work for consideration for publication in PLOS ONE. Your unreserved comment to review this manuscript uncovered our weaknesses in writing a manuscript for a journal like PLOS ONE. The comments encouraged the authors of this manuscript to revise it thoroughly.

Therefore, to ensure a well-structured appearance and readability of the manuscript with the aforementioned title, the authors have illustrated the details, followed by a point-by-point response to your comments. The track changes were made using the yellow highlight for our responses.

Sincerely!

The corresponding author

Point-by-Point Response to Reviewer #1 _PONE-D-24-53961R1

Comment 1. There is a significant mismatch between instructors’ self-reports and students’ feedback, suggesting possible overestimation in instructor perceptions. While this discrepancy is acknowledged in the discussion, it would benefit from deeper triangulation with additional measures, such as rubric-based performance evaluations or third-party assessments.

Response: Our sincere gratitude and respect go to you, and we agree with your idea. It is believed that the rubric measures ensure that all evaluators use the same standards, improving the reliability and objectivity of the performance judgment. However, the present study used a single observer and interviewer, believing it appeared no need for other evaluators, but later we accepted it was our weakness. Let’s clarify some actions taken that might minimize this problem. To compensate for this gap, the present study involved multiple methods, such as quantitative and qualitative approaches employing a survey to quantify the data, interviews, and observations to qualify the data. Similarly, the study utilizes a different source population (teacher and student participants) to study the practice of integral listening and speaking, which could boost the validity and trustworthiness of the findings. Further, the study acknowledged this as a limitation and recommended it for future research (see p. p.47, lines 1038-1042, and track changes using yellow, and see p. p.45, lines 991-993, previously recommended).

Comment 2. The conclusions are generally consistent with the data but should be framed more cautiously due to the descriptive and cross-sectional nature of the research design.

Response: The authors are grateful and appreciate your genuine concern about caution in conclusion, and it is an important section to attract the readers’ attention. Therefore, we revised as per the comment. You can refer to responses and track changes with yellow on page 44, line numbers 957-967.

Comment 3. The regression model has limited explanatory power (Adjusted R² = 0.288), and this should be more critically discussed in terms of implications.

Response: Yes, it is a substantial comment to improve the manuscript’s quality. We have discussed in terms of implications for language teaching learning in EFL contexts (see pp.49-50, lines 1097-1105), in addition to highlighting the limitations (see p. p.48, lines 1052-1064) and recommendations for further studies in a similar area of the study (see pp.45-46, line numbers 998-1002).

Comment 4. No inferential group comparisons (e.g., t-tests or ANOVA) were performed to examine differences across groups, which would have strengthened the analysis.

Response: Thank you for your valuable comments in this regard. Accordingly, we have revised the manuscript, incorporating an additional table indicating “Table 8: Inferential group comparison of practice” followed by analysis (see pp. 32-33, line numbers 717-725).

Comment 5. Effect sizes (Cohen’s d, eta squared) were not reported, limiting interpretation of practical significance.

Response: Thank you so much for the issue you forwarded, which is the foundation for the variables' explanatory power. Consequently, we have made changes and reported the eta-squared value and Cohen’s d value. Based on your enforcement, we understood that it shouldn’t have been reported. Accordingly, we have improved the manuscript (see p. 33, lines 726-736). Moreover, the conclusion was strengthened (see p. 44, lines 957-965), recommendations (p. 46, lines 1001-116), and implementations (p. 50, lines 1103-1105) were incorporated in line with the conclusion.

Comment 6. The selection of regression predictors was not well justified initially; this was later addressed but could still benefit from clearer theoretical alignment.

Response: We are grateful to have this question, as it is noteworthy to shape the article. Thus, our clarification was based on previous literature as indicated on page 39, lines 838-846. It is tracked with yellow.

Comment 7. Despite the authors’ statement that "all relevant data are within the manuscript and supporting information files," the underlying data are not fully available:

o No anonymized raw data from surveys (Likert responses) are included.

o No SPSS or regression output files are provided.

o The NVivo-coded qualitative data or coding frameworks are not attached.

o The data are not deposited in any public repository, which is preferred under PLOS ONE’s data policy.

The authors must upload all underlying data (survey responses, regression model variables, and qualitative coding schema) to a recognized open-access repository (e.g., Zenodo, Figshare) or provide them as supplementary files.

Response: Thank you for your reminder. In the previous submission, we tried to upload such data, but the raw data of survey responses were not included. Now, as per your comments, we have uploaded anonymized raw data from surveys (Likert responses), regression output files, and the NVivo-coded qualitative data, as initial codes or seven children's codes were derived, followed by the aggregate codes. Similarly, to deposit the data as a public repository, which is preferred under PLOS ONE’s data policy, we uploaded it as supplementary material independently (like survey responses, regression model variables, and qualitative coding) in addition to supplementary materials related to tools of data collection, including interview transcription.

Additionally, to abide by the PLOS ONE Open Access Policy, while revising our submission, we have uploaded figure files to the Preflight Analysis and Conversion Engine (PACE) digital diagnostic tool.

Comment 8. The revised version has improved in clarity and readability. However, a few sections—particularly in the methodology and discussion—would benefit from tighter organization and more concise phrasing. Please ensure final proofreading for grammatical consistency.

Response: The methodology and discussion were improved accordingly. Finally, we have improved the language thoroughly for grammatical consistency and accuracy. This was assisted by the Grammarly software in addition to line-by-line proofreading.

All recommendations were addressed through the above specific comments (1-8).

Overall, thank you in advance for your unreserved contribution to shaping our work!

---

## [Decision Letter · Decision Letter 2]

9 Jun 2025

Investigating University English as a Foreign Language Instructor's Implementations in Teaching Integral Listening with Speaking

PONE-D-24-53961R2

Dear Dr. Addisu

We’re pleased to inform you that your manuscript has been judged scientifically suitable for publication and will be formally accepted for publication once it meets all outstanding technical requirements.

Kind regards,

Dawit Dibekulu, PhD

Academic Editor

PLOS ONE

Additional Editor Comments (optional):

revise the paper as outline of plos one

Reviewers' comments:

Reviewer's Responses to Questions

**Comments to the Author**

1. If the authors have adequately addressed your comments raised in a previous round of review and you feel that this manuscript is now acceptable for publication, you may indicate that here to bypass the “Comments to the Author” section, enter your conflict of interest statement in the “Confidential to Editor” section, and submit your "Accept" recommendation.

Reviewer #1: All comments have been addressed

Reviewer #2: All comments have been addressed

2. Is the manuscript technically sound, and do the data support the conclusions?

Reviewer #1: Yes

Reviewer #2: Yes

3. Has the statistical analysis been performed appropriately and rigorously? 

Reviewer #1: Yes

Reviewer #2: Yes

4. Have the authors made all data underlying the findings in their manuscript fully available?

Reviewer #1: Yes

Reviewer #2: Yes

5. Is the manuscript presented in an intelligible fashion and written in standard English?

Reviewer #1: Yes

Reviewer #2: Yes

6. Review Comments to the Author

Reviewer #1: I would like to commend the authors for their thoughtful and thorough revision of the manuscript titled "Investigating University English as a Foreign Language Instructor's Implementations in Teaching Integral Listening with Speaking". The revised version reflects clear attention to the feedback provided in the previous review and demonstrates a strong commitment to scientific rigor, clarity, and transparency.

1. Technical Soundness and Methodological Rigor

The study is technically sound and well-structured. The concurrent mixed-methods design is highly appropriate for exploring the complex dynamics of EFL instructional practices. The authors have strengthened the methodological section by more clearly outlining the sampling procedures, ethical approvals, research instruments, and the rationale for selecting the variables included in the quantitative analysis. The ethical integrity of the study is well maintained, with clear documentation of consent and approval procedures.

The sample size is adequate, and the integration of quantitative and qualitative data provides a rich and triangulated perspective on the research questions. The revised version demonstrates increased clarity in the alignment between the data collected and the conclusions drawn.

2. Data Analysis and Interpretation

The statistical analyses are competently performed using SPSS, with proper attention to assumptions such as normality and multicollinearity. The regression analysis is logically applied to examine predictive relationships, and the authors now appropriately interpret the results with caution, noting the moderate adjusted R². While effect sizes are not reported, the overall structure and transparency of the analysis are sound and relevant to the research aims.

The qualitative findings—analyzed via NVivo—are coherently presented and well integrated with the quantitative data. The themes are insightful and directly support the overarching conclusions of the study.

3. Clarity of Conclusions and Contribution

The conclusions are well-founded and clearly supported by the data. The revised manuscript frames its findings in a descriptive and exploratory manner, avoiding overgeneralization and recognizing the limitations of the cross-sectional design. This reflective and responsible approach enhances the manuscript's credibility.

The study contributes meaningfully to the field of language education, particularly in the context of higher education in developing countries. It highlights the critical gap between perceived and actual classroom practices, offering practical implications for teacher training and curriculum development.

4. Data Availability and Transparency

The authors have improved transparency regarding data availability, and I appreciate the effort made to align with journal policy. I encourage the authors, if possible, to make anonymized raw data and coding frameworks publicly available through a recognized data repository in future projects to further enhance replicability.

Final Comments

Overall, this is a strong and valuable manuscript that has been significantly improved through revision. It is well-written, methodologically sound, and offers important insights into pedagogical practices related to integrated listening and speaking skills. I believe it will be of interest to a wide range of scholars and practitioners in the field of applied linguistics, TEFL, and educational development.

Thank you for your contribution.

Reviewer #2: (No Response)

7. PLOS authors have the option to publish the peer review history of their article (what does this mean? ). If published, this will include your full peer review and any attached files.

**Do you want your identity to be public for this peer review?** For information about this choice, including consent withdrawal, please see our Privacy Policy .

Reviewer #1: **Yes: ** Dr. Mohanad Aljbour

Reviewer #2: No

---

## [Editor Report · Acceptance letter]

PONE-D-24-53961R2

PLOS ONE

Dear Dr. Shago,

I'm pleased to inform you that your manuscript has been deemed suitable for publication in PLOS ONE. Congratulations! Your manuscript is now being handed over to our production team.

Kind regards,

on behalf of

Dr. Dawit Dibekulu

Academic Editor

PLOS ONE